# A miniaturized bionic ocean-battery mimicking the structure of marine microbial ecosystems

Huawei Zhu [1,2], Liru Xu[1,2], Guodong Luan[3], Tao Zhan [4,5], Zepeng Kang[4,5], Chunli Li[6], Xuefeng Lu [3] ✉, Xueli Zhang [4,5] ✉, Zhiguang Zhu [4,5] ✉, Yanping Zhang [1] ✉ & Yin Li [1] ✉

Marine microbial ecosystems can be viewed as a huge ocean-battery charged by solar energy. It provides a model for fabricating bio-solar cell, a bioelectrochemical system that converts light into electricity. Here, we fabricate a bio-solar cell consisting of a four-species microbial community by mimicking the ecological structure of marine microbial ecosystems. We demonstrate such ecological structure consisting of primary producer, primary degrader, and ultimate consumers is essential for achieving high power density and stability. Furthermore, the four-species microbial community is assembled into a spatial-temporally compacted cell using conductive hydrogel as a sediment-like anaerobic matrix, forming a miniaturized bionic ocean-battery. This battery directly converts light into electricity with a maximum power of 380 μW and stably operates for over one month. Reproducing the photoelectric conversion function of marine microbial ecosystems in this bionic battery overcomes the sluggish and network-like electron transfer, showing the biotechnological potential of synthetic microbial ecology.

The ocean, which covers ~70% of the Earth's surface, is a huge solar energy converter[1]. Approximately one-half of the global primary production occurs in the ocean[2–4]. It is estimated that 90% of marine biomass is microorganisms, which are centrally involved in the energy conversion[1,5–7]. In marine microbial ecosystems, the primary producers in the euphotic zone of water column such as the cyanobacteria *Prochlorococcus* and *Synechococcus* harvest solar energy through photosynthesis, fix carbon dioxide, and release organic matter (Fig. 1a)[8,9]. Organic matter can be consumed by heterotrophic plankton lived in the water column, or deposited into the marine sediments through sinking and burial[10]. Marine sediments is a large anaerobic bioreactor where organic matter is slowly degraded and fully oxidized by two types of heterotrophic microorganisms, eventually achieving complete remineralization (Fig. 1a)[11]. One type are the primary degraders, also known as fermentative microorganisms[12,13], which are responsible for anaerobic degradation of complex or high-molecular-weight organic compounds (e.g., polysaccharides and proteins) into smaller organic compounds (e.g., organic acids and amino acids)[14]. Another type are the ultimate consumers, which are responsible for the complete oxidation of organic compounds into carbon dioxide by respiring alternative electron acceptors, including $NO_3^-$, $Mn^{4+}$, $Fe^{3+}$ and $SO_4^{2-}$ (Fig. 1a)[15]. When all alternative electron acceptors were exhausted, methanogenesis by archaea becomes an important process for organic matter decomposition in deep sediments[16]. Through photosynthetic

[1]CAS Key Laboratory of Microbial Physiological and Metabolic Engineering, State Key Laboratory of Microbial Resources, Institute of Microbiology, Chinese Academy of Sciences, Beijing 100101, China. [2]University of Chinese Academy of Sciences, Beijing 100049, China. [3]Key Laboratory of Biofuels, Qingdao Institute of Bioenergy and Bioprocess Technology, Chinese Academy of Sciences, Qingdao 266101, China. [4]Tianjin Institute of Industrial Biotechnology, Chinese Academy of Sciences, Tianjin 300308, China. [5]National Center of Technology Innovation for Synthetic Biology, Tianjin 300308, China. [6]Institutional Center for Shared Technologies and Facilities, Institute of Microbiology, Chinese Academy of Sciences, Beijing 100101, China. ✉ e-mail: lvxf@qibebt.ac.cn; zhang_xl@tib.cas.cn; zhu_zg@tib.cas.cn; zhangyp@im.ac.cn; yli@im.ac.cn

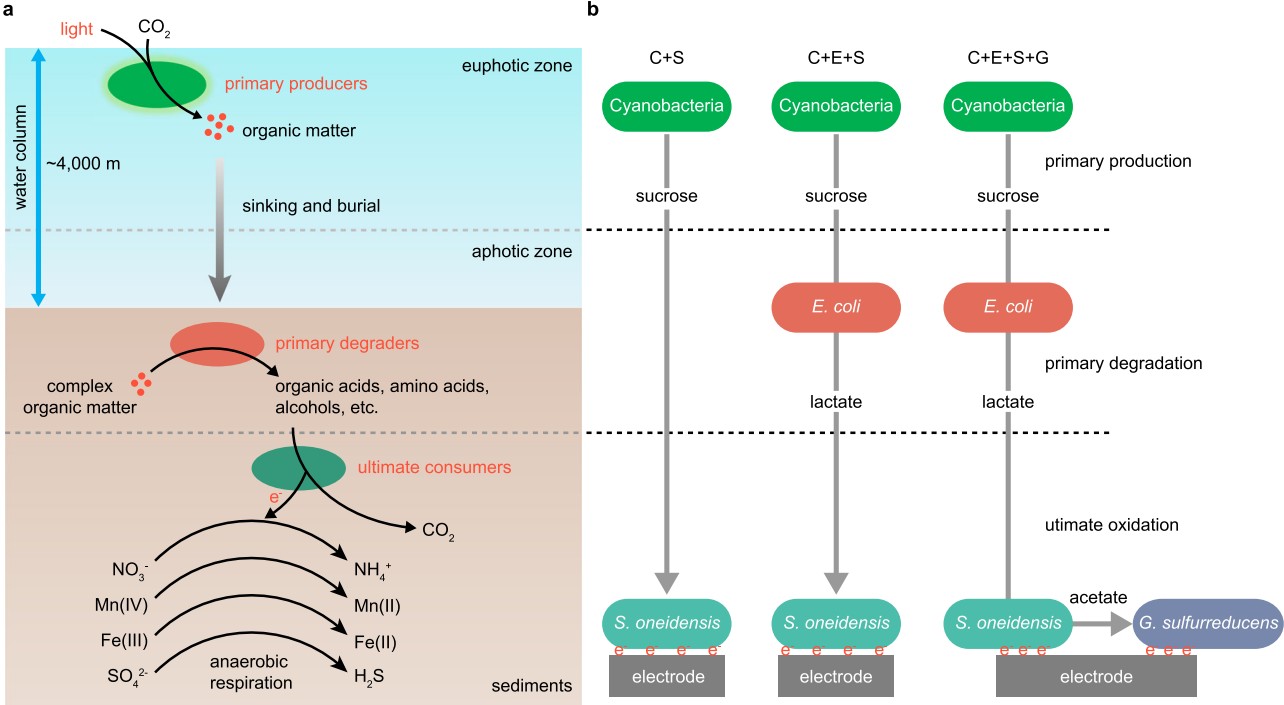

Marine microbial ecosystems | Synthetic microbial communities

**Fig. 1 | The structure of marine microbial ecosystems and synthetic microbial communities. a** Schematic representation of marine microbial ecosystems structured by primary producers, primary degraders and ultimate consumers. Primary producers in the euphotic zone absorb solar energy and store it in the form of organic matter. Primary degraders in sediments degrade complex organic compounds into small organic compounds. Ultimate consumers in deep sediments oxidize small organic compounds completely through anaerobic respiration with $NO_3^-$, $Mn^{4+}$, $Fe^{3+}$, $SO_4^{2-}$ as electron acceptors. During these processes, the electron flow supports microbial growth, metabolism and cycling of elements. **b** Three synthetic microbial communities, including the

two-species (C+S), three-species (C+E+S) and four-species (C+E+S+G) systems, were designed for photoelectric conversion. A sucrose-secreting strain of cyanobacteria was chosen as the primary producer, *E. coli* was chosen as the primary degrader, while *S. oneidensis* and *G. sulfurreducens* were used as the ultimate consumers. Electricity is generated by *S. oneidensis* and *G. sulfurreducens* through anaerobic respiration with an electrode as terminal electron acceptor. Among the three synthetic microbial communities, the four-species microbial community is the system completely mimicking the structure of the marine microbial ecosystems and forms the basis of the miniaturized bionic ocean-battery.

carbon fixation and remineralization of organic matter, the microbial communities that dominate the marine ecosystems drive biogeochemical cycles and solar energy conversion[1,13].

From the perspective of energy, marine microbial ecosystems can be viewed as a huge rechargeable battery charged by solar energy, in which the charging and discharging processes are cycled. In the charging process, the photosynthetic microorganisms in the ocean's surface use solar energy to fix carbon dioxide into organic matter. In the discharging process, the charged energy (in the form of chemical bonds) flows into different microbial species of the ecosystems via step-wise degradation of organic matter by heterotrophic microorganisms, eventually releasing carbon dioxide. These repeated charging/discharging cycles provide energy to all living organisms and sustain their lives in the marine microbial ecosystems. For this huge solar energy conversion system, we coin the term ocean-battery.

The majority of heterotrophic microorganisms in the marine microbial ecosystems reside in the upper sediment layers of the seafloor[10,12]. The average depth of the water column in the ocean is 4000 m, but photosynthesis only occurs in the top 200 m reached by sunlight (Fig. 1a)[12]. This means that the distance from the primary production to ultimate oxidation exceeds thousands of meters. Due to the large spatial scale of the charging and discharging processes of the ocean-battery, the organic matter deposited from the ocean's surface is recycled at a geological timescale of thousands of years[17–19]. Moreover, the ocean-battery is a highly intricate system where electrons are transferred among microbial species in a network-like way due to its vast microbial diversity and incredibly complex interspecies

interactions. This results in a capillary-like electron flow, which decreases the electron transfer efficiency, and makes targeted electron allocation difficult.

In this work, we aim to develop a bio-solar cell by mimicking the basic ecological structure of marine microbial ecosystems, but overcoming the low electron transfer efficiency resulted from the large spatial-temporal scales. To this end, we reproduce the photoelectric conversion function of the huge and complex ocean-battery in a compact and simple cell. Furthermore, the complex microbial community in the ocean is simplified so as to ensure directed and targeted electron flow. Specifically, we design a synthetic microbial community composed of specific microorganisms affiliated to three ecological niches, including primary production, primary degradation and ultimate oxidation. Using conductive hydrogel as a sediment-like anaerobic matrix, this synthetic microbial community is assembled into a miniaturized bionic ocean-battery with marine microbial ecological structure, which can stably convert light into electricity.

## Results

### Design of the synthetic microbial communities

A bio-solar cell reproducing the photoelectric conversion function of the ocean-battery must comprise specific microorganisms affiliated to the three niches of marine microbial ecosystems. Carbohydrates make up the largest fraction of marine organic matter[20], among which sucrose can be accumulated to a high concentration by many cyanobacteria, especially in environments with fluctuating salinity[21]. Therefore, the engineered *Synechococcus elongatus* strain Syn7942-FL130

(Supplementary Table 1)[22] capable of accumulating up to 3.0 g·L$^{-1}$ of extracellular sucrose, which accounts for 67.6% of total light energy fixed (Supplementary Fig. 1), was chosen as the primary producer. For the primary degrader, since the fermentative bacterium *Escherichia coli* was also found to persist autochthonously in environment matrices such as sediments, sands and soils[23–25], an engineered *E. coli* strain HX030-Suc (Supplementary Table 1) capable of fermenting sucrose into D-lactate under anaerobic conditions with a yield of over 85%, was chosen (Supplementary Fig. 2). *Shewanella oneidensis* MR-1 and *Geobacter sulfurreducens* PCA are two well-known dissimilatory metal reducing microorganisms. These two species are widely distributed in marine environments, especially in deep sediments[26,27]. *S. oneidensis* and *G. sulfurreducens* cannot utilize monosaccharides or polysaccharides, but they can utilize most of the acidic fermentation products of carbohydrates and transfer electrons outside the cells[27,28]. Under anaerobic conditions, *S. oneidensis* oxidizes lactate into acetate[29], while *G. sulfurreducens* is able to completely oxidize acetate into carbon dioxide[30]. Considering these features, both *S. oneidensis* and *G. sulfurreducens* were chosen as ultimate consumers, responsible for the complete oxidation of lactate and electricity generation. The complete synthetic microbial community is composed of four species, namely cyanobacteria (C), *E. coli* (E), *S. oneidensis* (S) and *G. sulfurreducens* (G), and termed as a four-species system (C + E + S + G) (Fig. 1b). For comparison, a three-species system (C + E + S) lacking the ultimate consumer G, and a two-species system (C + S) lacking the primary degrader E and the ultimate consumer G were included as controls (Fig. 1b). E and S were further engineered to meet different physiological needs. The genes encoding three nitrate reductases (NarG, NapA, NarZ) in *E. coli* strain HX030-Suc were knocked out to enable it to ferment sucrose in the nitrate-containing MBG11-S medium, and the resulting strain was designated as E2 and used in the subsequent experiments (Supplementary Table 1 and Supplementary Fig. 2). As *S. oneidensis* is not able to utilize sucrose, an expression cassette encoding sucrose catabolism genes was introduced into *S. oneidensis* (Supplementary Fig. 3), and the resulting strain S2 was used in the two-species system (C + S), specifically designated as (C + S2). Another *S. oneidensis* strain harboring an empty plasmid, designated as S1, was used in the three-species system (C + E + S) and four-species system (C + E + S + G), and specifically designated as (C + E2 + S1) and (C + E2 + S1 + G), respectively. The different engineered strains of E and S labeled with different subscripts are indicated in Supplementary Table 1 and the relevant figure legends.

## Constructing synthetic microbial communities attached to porous electrodes

Marine sediments provide an anaerobic and spatially structured environment for microbial colonization[12]. To provide a home for the synthetic microbial community, a thick porous electrode was used to enable the community to reside and function, thus forming a basic electrochemical component of a bio-solar cell (Fig. 2a). This porous electrode was called rectangular polypyrrole (RPPy), which was prepared by carbonizing the conductive polymer polypyrrole at a high temperature (Supplementary Fig. 4a)[31]. For comparison, the commercial carbon cloth with smaller thickness was used (Fig. 2c and Supplementary Fig. 4b). Meanwhile, to avoid the disturbance of oxygen produced by cyanobacteria to the anodic biofilm, the synthetic microbial communities attached to RPPy and carbon cloth were constructed using a mode of temporal separation organization, in which the charging process of photosynthetic sucrose production and the discharging process of anaerobic sucrose oxidation were implemented sequentially under light and dark conditions, respectively. The electrochemical systems were operated at two external loads, whereby low resistance corresponds to high current density. The RPPy-supported synthetic microbial communities showed a stable current density and a short acclimation period (Fig. 2d, f). This is in contrast to

the unstable current density and the long acclimation period when using carbon cloth (Fig. 2e, g). The obvious signal blunt changes shown in Fig. 2e, g indicated that the biofilm formed on carbon cloth was possibly the least stable, thus was most sensitive to fluctuations introduced by great change of electrical current during linear sweep voltammetry (LSV) scanning. Ideally, a three-layered spatial distribution of the four-species synthetic microbial community is expected to form to maximize the efficiency of electron transfer (Fig. 2b). We tried to observe the actual distribution of the bacteria but failed due to the interference of the black electrode. The porous structure of RPPy is likely to support faster and steadier biofilm formation than the carbon cloth does, suggesting that RPPy can be used to better support the spatial organization of the synthetic microbial community.

Among the three RPPy-supported synthetic microbial communities, the four-species system (C + E2 + S1 + G) always outperformed the other two in terms of current density, stability and longevity (Fig. 2d, f). Due to the complete oxidation of acetate by *G. sulfurreducens*, the total coulomb output of the four-species system (C + E2 + S1 + G) was considerably higher than that of the other two (Supplementary Fig. 5a). The relatively lower coulombic efficiency of the four-species system compared to the two or three-species systems (Supplementary Fig. 5b) was due to the theoretical coulomb output from sucrose to $CO_2$ (but not acetate) was taken into account. A comparison of the three-species system (C + E2 + S1) with the two-species system (C + S2) revealed that the presence of the primary degrader E improved and stabilized the current density (Fig. 2d, f). Although the strain S2 used in the two-species system (C + S2) can utilize sucrose directly, the anaerobic oxidation of sucrose was far slower than the oxidation of lactate (Supplementary Fig. 6). These results indicated that mimicking the basic structure of a natural ecosystem is critical for the synthetic microbial community to achieve the desired performance.

## Electrochemical characterization of the synthetic microbial communities

To further explore the photoelectric conversion potential of the RPPy-supported synthetic microbial communities, several electrochemical techniques were employed for in-depth characterization. Cyclic voltammetry (CV) was applied to reveal the kinetics of redox reactions at the cell-electrode interface. A typical redox peak of electroactive biofilms starting from around −0.4 V vs. Ag/AgCl appeared in the CV curves and the four-species system (C + E2 + S1 + G) showed the highest catalytic current (Fig. 3a and Supplementary Fig. 7a). Electrochemical impedance spectroscopy (EIS) was used to determine the interfacial charge-transfer resistance of the anode. The Nyquist plots showed the well-defined semicircles over the high frequency range (Fig. 3b and Supplementary Fig. 7b), and the diameter of semicircle corresponds to the charge-transfer resistance ($R_{ct}$). The $R_{ct}$ values of the two-, three-, and four-species systems ($R_{ct}$ = 175 Ω, 73 Ω, 44 Ω, respectively) with RPPy as anode were remarkably smaller than those of the systems ($R_{ct}$ = 1463 Ω, 1185 Ω, 779 Ω, respectively) with carbon cloth as anode, implying the faster electron transfer on the RPPy-biofilm interface. Meanwhile, the smallest $R_{ct}$ occurred in four-species system (C + E2 + S1 + G) indicated the hybrid electroactive biofilm of *S. oneidensis* and *G. sulfurreducens* was more conductive than the single biofilm of *S. oneidensis*. This might be ascribed to the enhanced electron transfer by synergetic oxidation of lactate and acetate.

The polarization curves and power output curves were used to further evaluate the performance of three synthetic microbial communities. As shown in Fig. 3c and Supplementary Fig. 7c, the four-species system (C + E2 + S1 + G) had a less steep decreasing slope and a higher maximum current density than the other two systems, suggesting a smaller internal resistance, which was in accordance with the EIS analysis. Additionally, among the RPPy-supported microbial communities, the maximum power density of the four-species system

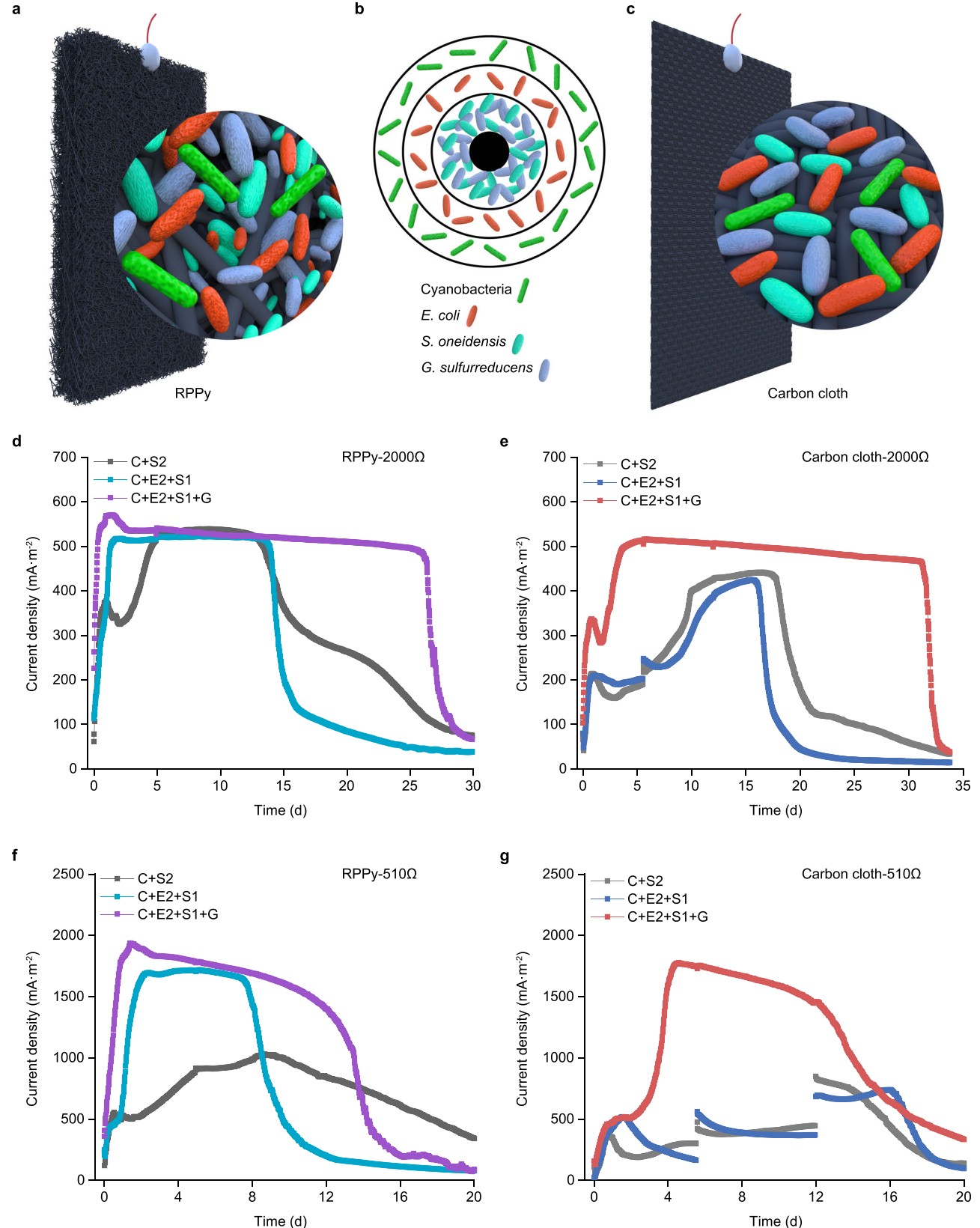

(C + E2 + S1 + G) reached up to 1.7 W·m⁻², which was 42% and 183% higher than that of the three-species system (C + E2 + S1) and two-species system (C + S2), respectively (Fig. 3d). The maximum power density of the corresponding systems using carbon cloth is all lower, among which the highest power density of 1.1 W·m⁻² was also obtained from the four-species system (C + E2 + S1 + G) (Supplementary Fig. 7d).

The same trends appeared in the systems using 2000 Ω external load for anode acclimation (Supplementary Fig. 8). The power overshoot phenomenon observed in Fig. 3d and Supplementary Fig. 8b might be due to that the demand for electrons of the external circuit at high current condition exceeds the electron generating rate of microorganisms[32]. The maximum power density of 1.7 W·m⁻² achieved

**Fig. 2 | Electricity generation of the synthetic microbial communities attached to porous electrodes. a** Illustration of the porous electrode of RPPy. RPPy is thick and porous, providing large space for microbial colonization. **b** A top view of the ideal three-layered structure of the four-species microbial community. **c** Illustration of carbon cloth. **d, f** Current density generated by the three synthetic microbial communities with RPPy as anode at external resistances of 2000 and 510 Ω, respectively. LSV scanning was conducted at day 5. **e, g** Current density generated by the three synthetic microbial communities with carbon cloth as anode at external resistances of 2000 and 510 Ω, respectively. The obvious signal fluctuations in panels **e, g** were caused by LSV scanning at days 5.5 and 12. The current

densities were normalized to the geometrical area of anode. Three synthetic microbial communities include a two-species system (C + S2), a three-species system (C + E2 + S1) and a four-species system (C + E2 + S1 + G), where C represents a sucrose-producing cyanobacterium, E2 represents an engineered *E. coli* capable of degrading sucrose to lactate in MBG11-S medium, S1 represents an engineered *S. oneidensis* capable of generating electricity by oxidizing lactate, S2 represents an engineered *S. oneidensis* capable of generating electricity by oxidizing sucrose directly, and G represents *G. sulfurreducens* capable of generating electricity by completely oxidizing acetate. Source data are provided as a Source Data file.

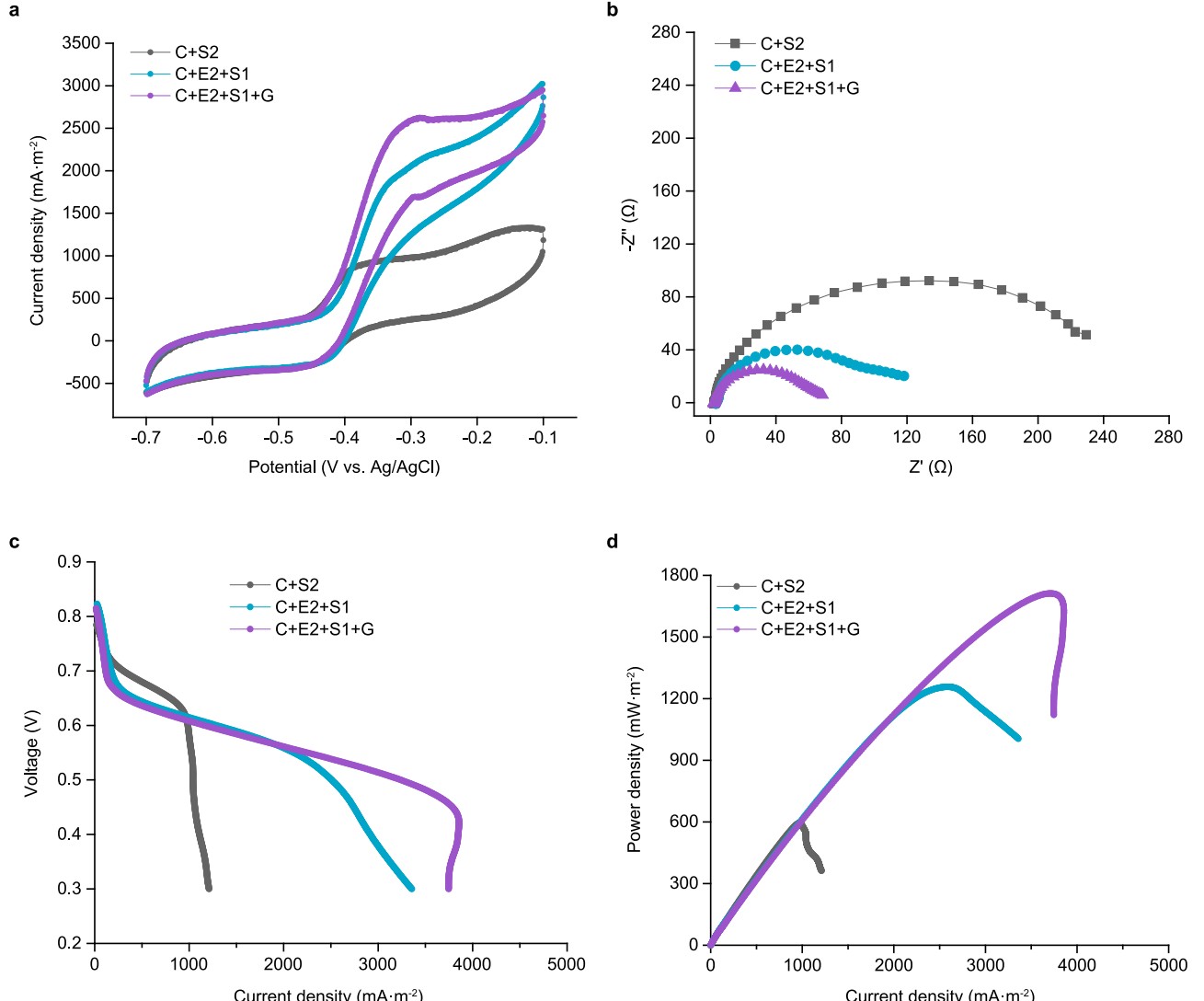

**Fig. 3 | Electrochemical characterization of three synthetic microbial communities attached to RPPy electrode. a** CV curves at a scan rate of 1 mV·s⁻¹. **b** Nyquist plots of the anode in a frequency range of 100 kHz to 1 mHz. **c** Polarization curves obtained from LSV. **d** Power curves derived from polarization

curves. The external resistance for anode acclimation before the measurement is 510 Ω. The current/power densities were normalized to the geometrical area of anode. The specific functions of strains C, E2, S1, S2 and G are described in the legend of Fig. 2. Source data are provided as a Source Data file.

in the RPPy-supported four-species microbial community exceeds those of previously reported bio-solar cells by more than two-fold (Supplementary Fig. 9a and Supplementary Data 1)[33]. The constant-current discharge was used to further evaluate the actual performance of the RPPy-supported synthetic microbial communities. The voltage of the four-species system (C + E2 + S1 + G) remained almost stable when the current rose from 0.1 to 1.0 mA, whereas the voltages of the three-species (C + E2 + S1) and two-species (C + S2) systems dropped rapidly once the current exceeded 0.6 and 0.1 mA, respectively (Supplementary Fig. 10). These results demonstrated that the RPPy-

supported four-species microbial community had great potential for photoelectric conversion.

**Characterizing the biofilms of synthetic microbial communities**
Microbial communities in natural sediments live in the form of biofilms[34]. Likewise, the synthetic microbial communities residing on electrodes may also form biofilms with different structures. Scanning electron microscopy (SEM) showed that biofilms were formed on the carbon fibers of the porous electrodes (Supplementary Fig. 11b, d). Since the carbon fibers are distributed loosely in RPPy but densely in

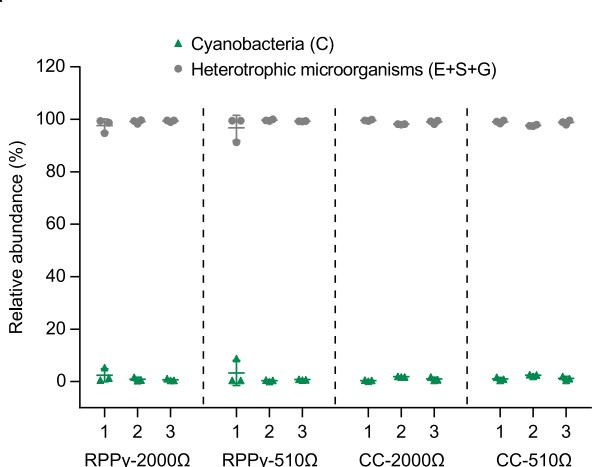

**Fig. 4 | Biofilm composition of the synthetic microbial communities. a** The relative abundance of cyanobacteria and heterotrophic microorganisms on the anodic biofilms. **b** The relative abundance of three heterotrophic microorganisms on the anodic biofilms. The labels 1, 2, and 3 on the x-axis represent C + S2, C + E2 + S1 and C + E2 + S1 + G, respectively. Abbreviation: CC carbon cloth. Data are presented as mean values ± SD from $n = 3$ independent experiments. Source data are provided as a Source Data file.

carbon cloth (Supplementary Fig. 11), RPPy may provide a larger space for the microbial communities to colonize. The total amount of biomass in the biofilms (calculated as the DNA content per unit area of electrode) indicated that the RPPy indeed harbored more cells than the carbon cloth (Supplementary Fig. 12). The composition of biofilms formed by the three microbial communities revealed that heterotrophic microorganisms were the dominant species, whereas only 1-3% of the biofilms were occupied by cyanobacteria (Fig. 4a). The relative abundance of *S. oneidensis* in the three-species systems (C + E2 + S1) was about 30-60% (Fig. 4b). Strikingly, when *G. sulfurreducens* was introduced in the four-species system (C + E2 + S1 + G), the relative abundance of *S. oneidensis* decreased sharply to 1% (Fig. 4b). Previous studies also found that *G. sulfurreducens* dominated the anodic biofilm when co-cultured with *S. oneidensis*, while the latter is found in the form of planktonic cells[35,36]. Different from *S. oneidensis*, the relative abundance of *E. coli* was not obviously decreased when *G. sulfurreducens* was introduced, and even increased in low resistor systems (Fig. 4b). This suggests the *S. oneidensis* and *G. sulfurreducens* may compete for the anode surface as they are both ultimate consumers, whereas *E. coli* as a primary degrader does not engage in direct competition with the ultimate consumers regardless of their composition.

### Fabricating a miniaturized bionic ocean-battery using conductive hydrogel matrix

The porous electrode provides a large space for the synthetic microbial community to colonize, but it cannot create an anaerobic environment for the primary degrader and the ultimate consumers to fully function. The dissolved oxygen gradient is a crucial factor that shapes the spatial distribution of marine microbial ecosystems[37]. In the upper ocean, primary producers produce massive amounts of oxygen during photosynthesis. The dissolved oxygen concentration in the ocean decreased gradually with increasing depth, mainly due to the weakened photosynthesis and the continuous consumption of oxygen through respiration[37]. Benefiting from the large spatial scale, the oxygen level in marine sediments falls to anoxic status, which enables resident microorganisms to carry out anaerobic metabolism. However, in a spatial-temporally compacted synthetic microbial community, the high concentration of oxygen generated through photosynthesis can be devastating for anaerobic heterotrophic microorganisms. Therefore, an apparent contradiction between photosynthetic oxygen evolution and anaerobic energy harvesting needs to be resolved. To this end, we encapsulated the heterotrophic microorganisms in a

conductive hydrogel, in which a relatively anaerobic environment can be formed (Fig. 5a). Besides conductivity, the conductive hydrogel allows mass transfer of sucrose, lactate and acetate, but keeps low oxygen permeability (Supplementary Fig. 13). Among different conductive materials, the PPy-based conductive hydrogel performed best in terms of electron collection (Supplementary Fig. 14). The current generated in the PPy conductive hydrogel was due to lactate oxidation by encapsulated cells of *S. oneidensis*, which excluded the contribution of the conductive hydrogel itself (Fig. 5b). These results indicated that a conductive hydrogel can be used to encapsulate microorganisms and thereby circumvent the negative effect of oxygen on electricity generation.

Conductive hydrogels can encapsulate cells, but cannot prevent some of the cells from escaping from the hydrogels. Since *E. coli* and *S. oneidensis* are facultative anaerobes and oxygen is available outside the hydrogels, the escaped cells may dissipate electrons through aerobic respiration, thus decreasing the overall energy conversion efficiency (Fig. 5a). To avoid this, the aerobic respiration chains of *E. coli* and *S. oneidensis* were blocked by deleting all genes encoding terminal oxidases (Fig. 5a). The resulting terminal oxidases-null mutant of *E. coli* (E3) could ferment sucrose into lactate under aerobic conditions with the same yield as under anaerobic conditions, even though its growth was inhibited (Fig. 5c and Supplementary Fig. 15). Thus, the escape of E3, if any, would not affect the overall energy conversion efficiency. The terminal oxidases-null mutant of *S. oneidensis* (S3) was able to grow under anaerobic conditions, but could not grow under aerobic conditions because of the production of intracellular reactive oxygen species (ROS) (Fig. 5d and Supplementary Fig. 16). In this case, strain S3 would die if escaped from the hydrogel. In short, after knocking out all the terminal oxidase genes, neither *E. coli* (E3) nor *S. oneidensis* (S3) could carry out aerobic respiration, while their anaerobic growth and metabolism were not affected.

Subsequently, two heterotrophic microorganisms (E3 + S3) that were rendered incapable of aerobic respiration, plus the strict anaerobe *G. sulfurreducens*, were encapsulated in a PPy conductive hydrogel, forming a sediment-like layer (Fig. 6a). The cyanobacteria strain of Syn7942-FL130 was cultivated in MBG11-S medium on the top of the conductive hydrogel, mimicking the oceanic water column (Fig. 6a). The distance between the top of the water column-like layer and the bottom of the sediment-like layer was only 5 cm. When the conductive hydrogel was used as the anode and paired with an air cathode, the whole synthetic microbial community was assembled into an all-in-one

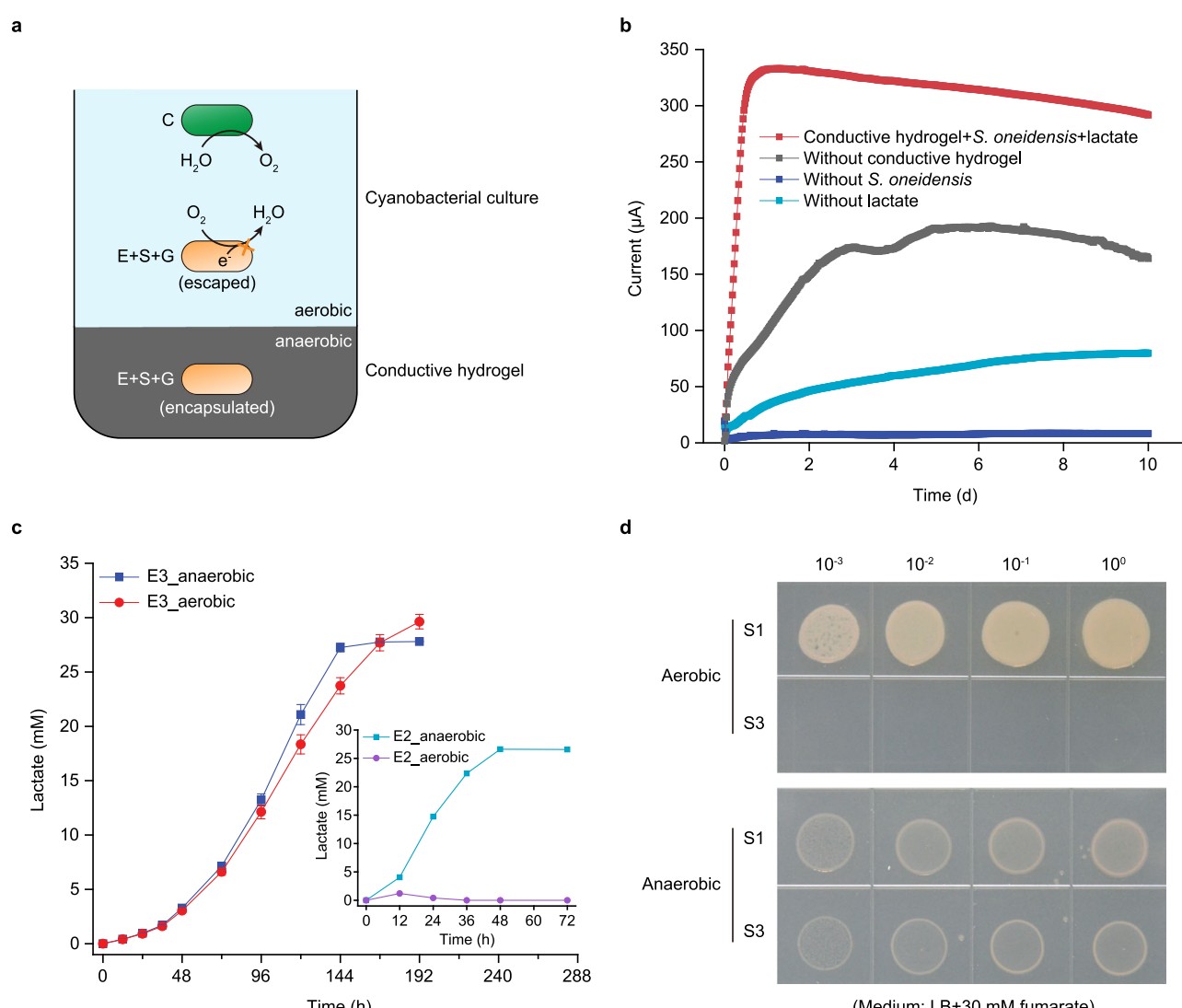

**Fig. 5 | Resolving oxygen contradiction by combining conductive hydrogel with aerobic respiration-null mutants. a** Schematic illustration of conductive hydrogel-based system. Three heterotrophic microorganisms (E + S + G) were encapsulated in a conductive hydrogel, in which a relatively anaerobic environment could be formed. To avoid the escaped cells dissipating electrons through aerobic respiration, the aerobic respiration chains of heterotrophic microorganisms were blocked. **b** Electricity generated by encapsulated *S. oneidensis* in PPy conductive hydrogel. Control groups without electron donor, without *S. oneidensis*, or without conductive hydrogel were included for comparison. **c** The fermentation of sucrose to lactate by the engineered *E. coli* strains E2 and E3 under anaerobic and aerobic conditions. Strain E3 is an aerobic respiration-null mutant. Data are presented as mean values ± SD from *n* = 3 independent experiments. **d** Drop plate test showing the growth defect of an aerobic respiration-null mutant of *S. oneidensis* (S3) under aerobic conditions. Source data are provided as a Source Data file.

bio-solar cell on a spatial-temporally compacted scale (Fig. 6a). We view this bio-solar cell as a miniaturized bionic ocean-battery, as it possesses not only the basic physical structure (water column layer and sediment layer) but also the basic ecological structure (primary producer, primary degrader, and ultimate consumer) of the ocean. Most importantly, it reproduces the ocean-battery's function of photoelectric conversion. The miniaturized bionic ocean-battery generated an electrical current of about 35 mA·m$^{-2}$ under light condition and maintained stably for over 30 days, whereas no significant current was generated in the dark (Fig. 6b). Likewise, there was no significant current generated when the cyanobacteria or *E. coli* was absent (Fig. 6b), the current peak during the early 2 days could be ascribed to the residual metabolites or a bit of organic matter introduced by the medium. Moreover, the current output showed a positive light-response under light/dark cycles (Supplementary Fig. 17). Usually, a direct photoelectric conversion system would show a sharp response upon on/off of the light. Since the bionic battery constructed here is a photon − energy carriers − electricity conversion system, it was

conceivable the observed light-response of our system (Supplementary Fig. 17) is not as sharp as that of the direct photoelectric conversion system. More importantly, the current density under light/dark cycles at the tenth day could reach the same level of that under continuous light, and maintained a relative stable output afterwards (Supplementary Fig. 17). This suggests our system might work under natural diurnal conditions. Collectively, these results demonstrated that the electricity generated by the miniaturized bionic ocean-battery was from light.

The polarization curve and power curve showed the maximum power output of a single miniaturized bionic ocean-battery was 380 µW (Fig. 6c). The overshoot phenomenon in polarization curve might also be an indication of mass-transfer limitation of sucrose, lactate, and acetate in conductive hydrogel. The EIS analysis of conductive hydrogel showed a small mass-transfer resistance appeared in low-frequency region, but the charge-transfer resistance is small ($R_{ct}$ = 60 Ω; Supplementary Fig. 18). According to the energy flux of each sub-process, the overall energy efficiency from light to electrical

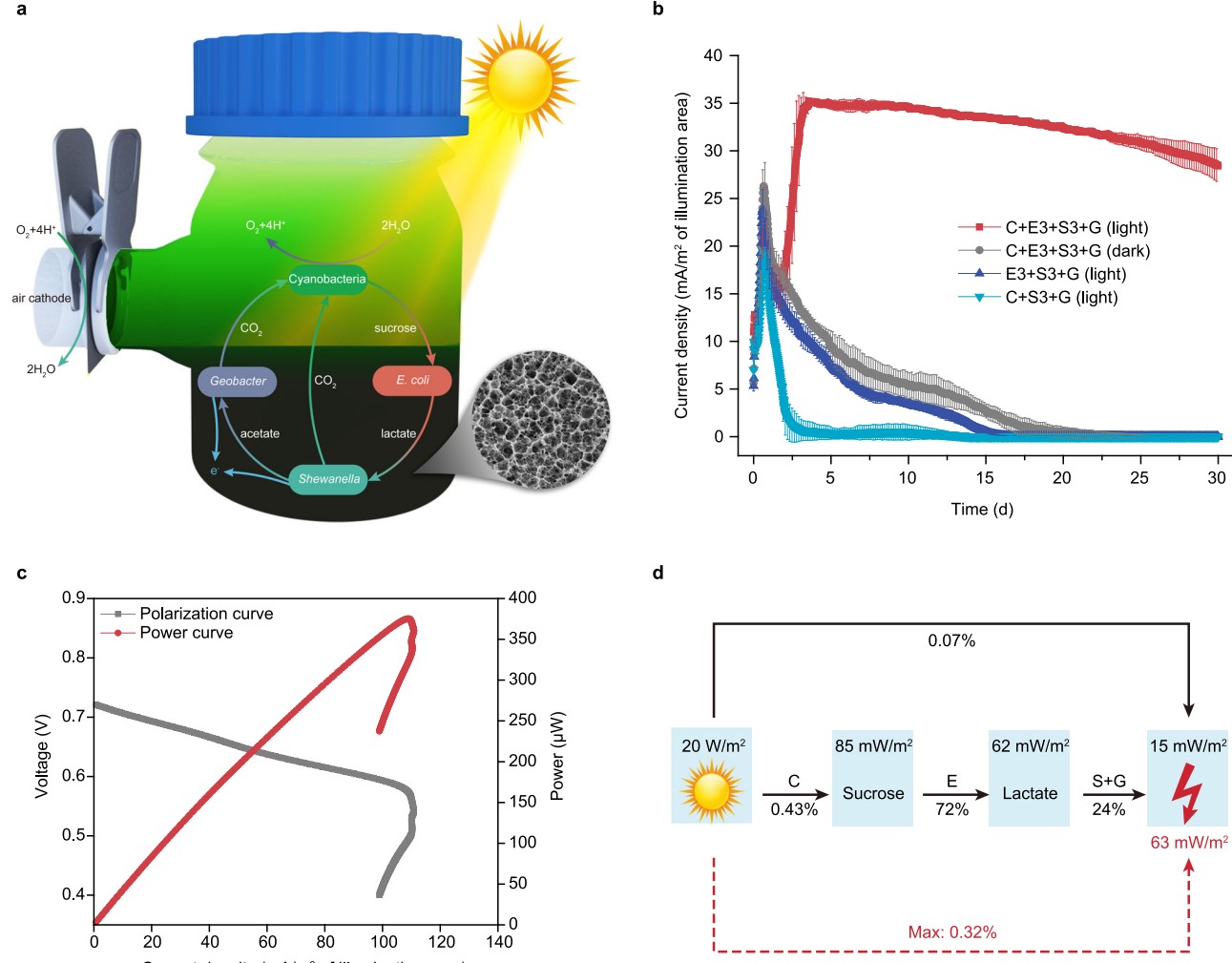

**Fig. 6 | Fabricating the miniaturized bionic ocean-battery using conductive hydrogel matrix. a** The diagram of a miniaturized bionic ocean-battery fabricated in a single-chamber electrochemical device. The bottom is PPy conductive hydrogel encapsulating the strains E3, S3 and G. The upper layer is cyanobacteria culture of strain Syn7942-FL130. The four-species community forms a complete carbon cycle ($CO_2$-sucrose-lactate-acetate-$CO_2$). In this unidirectional electron flow, the energy is inputted in the form of light and outputted in the form of electricity. The SEM image shows the porous structure of PPy conductive hydrogel. The air cathode is shown on the left. **b** Electricity generation of the miniaturized bionic ocean-battery powered by

light directly. The systems running in the dark, the systems free of cyanobacteria, and the systems free of *E. coli* were included as negative controls. The light source is white light-emitting diodes with intensity of 20 W·m⁻². Data are presented as mean values ± SD from *n* = 3 independent experiments. **c** Polarization curve and power curve of the miniaturized bionic ocean-battery. **d** The overall energy balance of the miniaturized bionic ocean-battery. The energy flux and energy efficiency of light-to-sucrose, sucrose-to-lactate, lactate-to-electricity and light-to-electricity are presented. The current/power densities were normalized to the geometrical area of receiving the light. Source data are provided as a Source Data file.

current under the external load of 2 kΩ was 0.07% (Fig. 6d). When considering the maximum power output, the overall energy efficiency could reach up to 0.32% (Fig. 6d). The overall energy efficiency was mainly limited by the process of light-to-sucrose conversion by cyanobacteria. These results indicated that the microbial communities composed of microorganisms that naturally reside kilometers apart, and with entirely different physiological characteristics, could be functionally reconstructed in a centimeter-scale device for photoelectric conversion.

## Discussion

In this study, the solar energy conversion function of the marine microbial ecosystems was modeled in a species-minimized and structurally simplified synthetic microbial community (Fig. 7). The synthetic microbial community consists of a primary producer, a primary degrader, and two ultimate consumers, which are responsible for photosynthetic electron storage in the form of organic matter, primary degradation of organic matter, and electricity generation, respectively.

This synthetic microbial community was designed following the microbial organization principles of marine microbial ecosystems. However, it is superior to the marine microbial ecosystems in terms of the unidirectional electron flow, where two adjacent microorganisms across different trophic levels were interconnected by a specific energy carrier. From complex to simplified, from disordered to ordered, this cascade organization enables energy convergence for maximal targeted electricity generation (Fig. 7). As such, the four-species synthetic community genetated a maximum power density of 1.7 W·m⁻² on a porous electrode, which was comparable to the power density achieved by a regular microbial fuel cell[38]. This power density is ten times higher than that of the ocean-battery (0.169 W·m⁻²), which was calculated by assuming the energy stored in organic matter in marine ecosystems was released in the form of heat (refer to Methods). By exploiting a conductive hydrogel to mimic the anaerobic sediment environment, the synthetic microbial community was assembled on a spatial-temporally compacted scale, which enabled direct electricity generation from light for over 30 days (Fig. 6b). The power output of

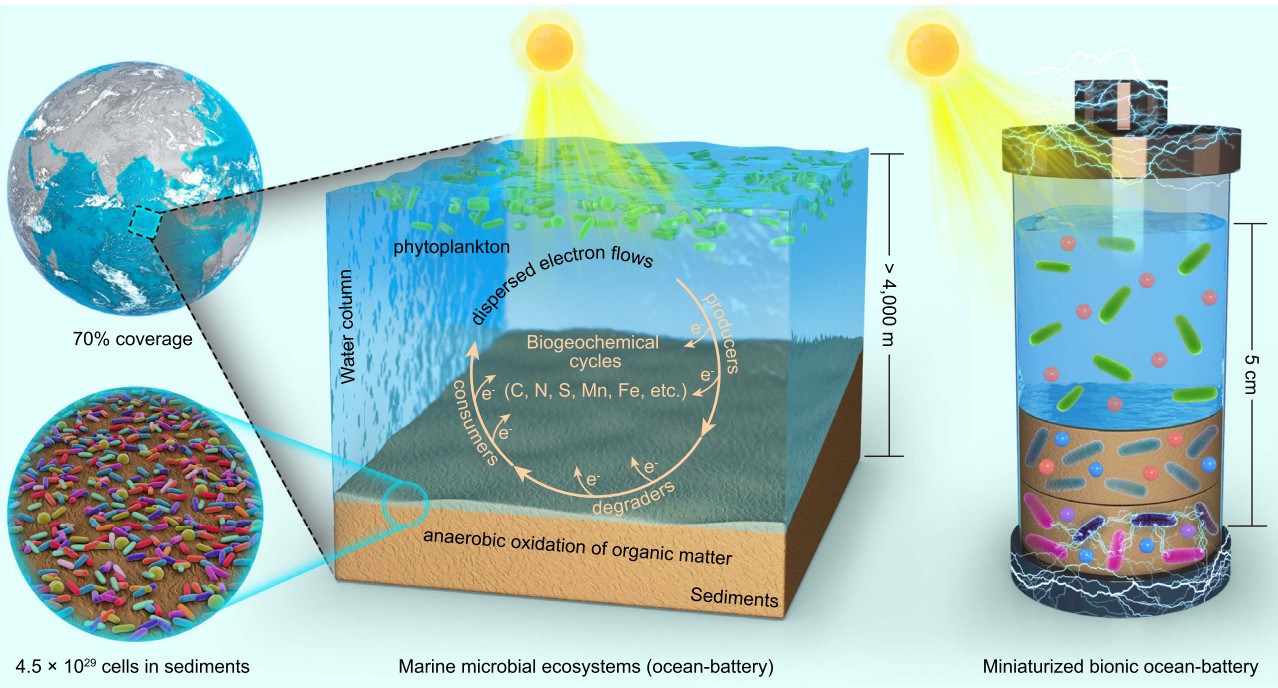

**Fig. 7 | The structure comparison of the marine microbial ecosystems and the miniaturized bionic ocean-battery.** Both systems possess same physical structure (water column layer and sediment layer) and same ecological structure (primary producers, primary degraders, and ultimate consumers). The marine microbial ecosystems are huge with the average depth exceeding 4000 m, while the miniaturized bionic ocean-battery was compacted in a vessel with a depth of 5 cm, thus accelerating the electron flow by shortening the electron transfer distance. In marine microbial ecosystems, especially in anaerobic sediments, the highly diversified microbial species and their complex interactions make the electron flow dispersed to various microbially mediated biogeochemical processes, i.e., elemental cycles. In contrast, the miniaturized bionic ocean-battery fabricated using the synthetic community only contains four microbial species connected by the specific energy carriers. This simplified structure targetedly directs electrons towards the only target, i.e., electrical current.

380 μW obtained in this study exceeded previously reported bio-solar cells, in which the highest value was 190 μW (Supplementary Fig. 9b and Supplementary Data 1)[39]. Therefore, our work presents an efficient miniaturized bionic ocean-battery with a minimized species number while retaining the full photoelectric conversion function of marine microbial ecosystems.

In this miniaturized bionic ocean-battery, the four microbial species with different functions worked in a cascade way to accomplish photoelectric conversion. Among these microorganisms, the primary degraders serve as a connecting link between primary producers and ultimate consumers. In particular, the primary degraders play an important role in providing accessible substrate for ultimate consumers, which was observed in many natural habitats[40–43]. In marine sediments, the fermentation products are the direct electron donors for terminal oxidation processes, such as iron and sulfur reduction, rather than complex organic matter[43–46]. In the synthetic microbial community that we constructed, the primary degrader *E. coli* breaks down sucrose into lactate and provides electron donor for *S. oneidensis*. Compared with sucrose catabolism, lactate oxidation is faster and more stable due to a shorter metabolic pathway, fewer rate-limiting steps and simpler regulatory network[47]. A complete system with microorganisms affiliated to three ecological niches also guaranteed maximum conversion of stored energy from primary production, and approximately two thirds of the solar energy fixed by the primary producer was converted into electricity (Supplementary Fig. 1). This energy reservoir was greatly enlarged compared with our previous system, which was able to convert only about 10% of the captured energy[48]. A better performance can be envisioned if the biofilm structure of the four-species synthetic community is further optimized.

The thick porous electrode of RPPy not only serves as an excellent electron collector, but also provides a suitable habitat for the multi-species synthetic community. The large specific surface area of RPPy is beneficial for the dense electrical wiring with ultimate consumers (*S. oneidensis* and *G. sulfurreducens*) for direct electron transfer. In addition to the electrical connection, the large inner space of RPPy ensures that the primary degrader is positioned in close proximity to the ultimate consumers, making intimate interactions among different species easier. Furthermore, the porous architecture also ensures effective diffusion of nutrients and dissipation of wastes for all species. Therefore, in the synthetic microbial community supported by the RPPy electrode, different species colonize spatially differentiated niches and exert their respective functions in harmony. On a thin carbon cloth, all species might fight for territory and eventually form a compact biofilm with sluggish electron transfer and inefficient mass transfer. Overall, the RPPy electrode reduces the distance between the primary degrader, the ultimate consumer, and the terminal electron acceptor, resulting in superior bioelectrochemical performance.

The pioneering usage of a conductive hydrogel makes it possible to organize the whole synthetic microbial community on a spatial-temporally compacted scale. The oxygen produced by the primary producer cannot penetrate easily into the conductive hydrogel due to large mass transfer resistance at the solid-liquid interface. Therefore, the conductive hydrogel created a sediment-like anaerobic environment for the primary degrader and ultimate consumers. By using the aerobic respiration-null strains, the risk of energy loss resulting from aerobic respiration could be avoided. The well untangled Gordian knot of oxygen enables the four species affiliated to three different ecological niches to effectively work together in a compacted space. From the large spatial scale of thousands of meters to a small spatial scale of a few centimeters, and from the long timescale of thousands of years to the short timescale of several days, this drastic compaction of spatial-temporal scales effectively accelerated the electron flow, thus overcoming the low electron transfer efficiency resulting from the

large spatial-temporal scales of the ocean-battery (Fig. 7). In short, the conductive hydrogel greatly shortened the long distance between primary production and anaerobic oxidation, thus reproducing the photoelectric conversion function of the huge ocean-battery in a small cell.

In view of the rechargeable and theoretically self-reproducing characteristics, it is possible that this miniaturized bionic ocean-battery may be expanded to a complex microbial ecosystem. In principle, the miniaturized bionic ocean-battery can be run independently in a closed prototype with solar energy as the sole input, because carbon is recycled after complete oxidation, and the water lost in photolysis is recovered at the cathode (Fig. 6a). On a terrestrial planet where solar irradiation is available, this bionic battery has the potential to operate, as long as water, carbon dioxide and minerals are available, as is the case on Mars[49–52]. Presently, the miniaturized bionic ocean-battery could run about one month, followed by collapse which was mainly caused by the death of cyanobacteria due to the deficiency of minimal inorganic nutrients. Thus, exploring the solutions to improve the sustainability of this system is needed.

Different from the common bio-solar cells developed based on individual photosynthetic microorganisms, the miniaturized bionic ocean-battery has the capacity of electron storage. This feature gives the miniaturized bionic ocean-battery flexibility in discharging, namely the electrons stored in organic matter are discharged only when electrical current is required, rather than discharging along with charging during the day. As for the potential applications of the miniaturized bionic ocean-battery, it is firstly hoped to serve as an alternative electrical energy source for ultralow-power facilities, such as environmental sensors of Internet of Things (IoT). Since the IoT sensors consume power in the range of µW to mW, a single miniaturized bionic ocean-battery generating hundreds of µW is sufficient to support these small facilities. Recently, a breakthrough was achieved by using a bio-solar cell to power a microprocessor continuously for half year[53]. Moreover, similar to the widely used solar cell, series-parallel stacking of bio-solar cells is one of the strategies to obtain the desired voltage and current output to power higher-power facilities. Installing bio-solar arrays on the building roofs or floating on the sea can also be envisioned. Furthermore, the large-scale fabrication of bio-solar cells is feasible by using 3D printing technology, especially for conductive hydrogel-based ones. Nevertheless, the practical output and robustness of such a bio-solar cell in natural environments remains to be investigated. Taken together, this study provides a model for the development of bio-solar cells based on synthetic microbial communities. The approach of extracting and reconstructing the basic structure of microbial communities opens up a way for exploiting the biotechnological potential of natural microbial ecosystems.

## Methods
### Strains and genetic manipulations
All strains used in this study are listed in Supplementary Table 1. The plasmids and primers used are listed in Supplementary Table 2 and Supplementary Data 2, respectively. Dlac-006 is a recombinant *E. coli* strain constructed for D-lactate production by deletion of the *pflB* gene encoding pyruvate formate lyase, the *frdABCD* operon encoding fumarate reductases, and the *mgsA* gene encoding methylglyoxal synthase, so that D-lactate was the only product when fermented under anaerobic conditions[54]. In this study, strain Dlac-006 was further modified for sucrose fermentation by integrating the sucrose utilization pathway. The *cscB* gene encoding sucrose permease (GenBank No. ADT76024.1) was amplified from the genomic DNA of *E. coli* W[55], while the *gtfA* gene encoding sucrose phosphorylase from *Bifidobacterium adolescentis* (GenBank No. WP_011742626.1)[56] was codon optimized and synthesized by GenScript Biotech Corporation. Subsequently, an artificial sucrose utilization operon was assembled by fusion PCR and

integrated into the chromosome using a two-step homologous recombination method[57,58]. The resulting strain was designated as HX030-Suc, in which the transcription of the *cscB* and *gtfA* genes was placed under the control of two strong constitutive promoters M1-93[59] and rrnB, respectively.

The CRISPR-Cas9 method[60] was used for gene knockout in *E. coli* HX030-Suc. Three nitrate reductases genes (*narG*, *napA*, *narZ*), three terminal cytochrome oxidases genes (*cyoABCD*, *appBC*, *cydAB*), and a quinol monooxygenase gene (*ygiN*) were knocked out successively. Briefly, a pTargetF-derived plasmid with a designed N20 sequence of the targeted gene and the corresponding homologous fragment were introduced into competent cells carrying the pCas plasmid, which expresses a Cas9 protein and a Red recombinase. The mutants were screened in Luria-Bertani (LB) agar medium containing 50 mg·L⁻¹ kanamycin and 50 mg·L⁻¹ spectinomycin at 30 °C, and verified by colony PCR and DNA sequencing. The pTargetF plasmid was cured by IPTG induction of pCas, and the temperature-sensitive pCas plasmid was cured by cultivating the strains at 37 °C. Taking the *narG* deletion as an example, the plasmid pTargetF-narG was obtained by self-ligation of the inverse PCR product amplified using the primer pair pTargetF-narG-F/pTargetF-R. The homologous fragment was formed by fusing two arms amplified using the primer pairs narG-up-F/narG-up-R and narG-down-F/narG-down-R. Subsequently, pTargetF-narG and the homologous fragment were electroporated into strain *E. coli* HX030-Suc (pCas).

A two-step homologous recombination method was used for gene knockout in *S. oneidensis*[61]. The up- and downstream fragments flanking the SO4606-4609 locus (*cox* gene) were amplified using the primer pairs SO4606-4609-5O/SO4606-4609-5I and SO4606-4609-3I/SO4606-4609-3O, respectively. The resulting two fragments were fused by overlap extension PCR using the primer pair SO4606-4609-5O/SO4606-4609-3O. The plasmid pRE112 was linearized by PCR using the primer pair pRE112-F/pRE112-R. The fused fragment was inserted into pRE112 using a Gibson assembly kit (New England BioLabs). The resulting plasmid pRE112-SO4606-4609 was first introduced into the plasmid donor strain *E. coli* WM3064 (an auxotroph, requires an external supply of 0.3 mM diaminopimelic acid for growth), and then transferred into *S. oneidensis*-ΔnapA by conjugation[62]. The primary mutants were selected by plating on LB medium supplemented with 30 mg·L⁻¹ chloramphenicol. Selection for second homologous recombination to remove the plasmid was accomplished by growing the primary mutants in LB medium without chloramphenicol for 12 h, followed by screening using LB agar medium containing 10% sucrose. The mutants were verified by colony PCR and DNA sequencing. The gene clusters of other two terminal oxidases, SO2361-2364 (*cco* gene) and SO3284-3286 (*cyd* gene), were successively deleted following the same procedure.

The DNA sequences (Supplementary Data 3) of the sucrose catabolism genes *glk* (encoding glucokinase), *cscA* (encoding sucrose hydrolase), *cscK* (encoding fructokinase), and *cscB* (encoding sucrose permease) from *E. coli* W were extracted from the KEGG database, and chemically synthesized by GenScript Biotech Corporation. The synthesized DNA sequences were respectively inserted into the expression plasmid pBBR1MCS-2 using a Gibson assembly kit. The plasmid pBBR1MCS-2 was linearized by PCR using the primer pair pBBR1MCS2-F/pBBR1MCS2-R, and two gene cassettes were amplified using the primer pair glk-cscAKB-F/glk-cscAKB-R. The resulting plasmids, pBBR1-glk-cscB and pBBR1-glk-cscAKB, were introduced into *E. coli* WM3064 and then transferred into *S. oneidensis*-ΔnapA by conjugation.

### Culture conditions
*Synechococcus elongatus* strain Syn7942-FL130[22] was routinely cultivated in BG11 medium (Supplementary Table 3) in an illumination incubator at 30 °C and 150 rpm, with 3% $CO_2$ and a light intensity of

$20\,W\cdot m^{-2}$. The light source of this incubator was white fluorescent lamps with full spectrum. For photosynthetic production of sucrose, the Syn7942-FL130 cells precultured in BG11 were centrifuged, washed and suspended in 50 mL MBG11-S medium (Supplementary Table 3) supplemented with 50 mM $NaHCO_3$ and 2 µM IPTG in 100-mL flasks. The initial $OD_{730}$ was adjusted to 0.6 and the other conditions remained unchanged.

$E.\ coli$ and $S.\ oneidensis$ strains were cultivated aerobically at 30 °C and 200 rpm in LB medium ($10\,g\cdot L^{-1}$ tryptone, $5\,g\cdot L^{-1}$ yeast extract and $10\,g\cdot L^{-1}$ NaCl) supplemented with $50\,mg\cdot L^{-1}$ kanamycin, $50\,mg\cdot L^{-1}$ spectinomycin or $30\,mg\cdot L^{-1}$ chloramphenicol where appropriate. The terminal oxidases-null $E.\ coli$ mutant E3 (Supplementary Table 1) was cultivated anaerobically in TB medium ($2.31\,g\cdot L^{-1}$ $KH_2PO_4$, $16.43\,g\cdot L^{-1}$ $K_2HPO_4\cdot3H_2O$, $12\,g\cdot L^{-1}$ tryptone, $24\,g\cdot L^{-1}$ yeast extract, $4\,mL\cdot L^{-1}$ glycerol and $3\,g\cdot L^{-1}$ glucose) in sealed bottles. The terminal oxidases-null $S.\ oneidensis$ mutant S3 (Supplementary Table 1) was cultivated anaerobically in LB medium supplemented with 30 mM sodium fumarate as electron acceptor in sealed bottles. The medium in sealed bottles was flushed with $N_2$-$CO_2$ (80:20) to remove oxygen before autoclaving.

$G.\ sulfurreducens$ PCA (DSM 12127), purchased from DSMZ, was cultivated anaerobically at 30 °C in ATCC 1957 $Geobacter$ medium[63]. The medium contained (per liter): 1.5 g $NH_4Cl$, 0.78 g $NaH_2PO_4\cdot2H_2O$, 0.1 g KCl, 2.5 g $NaHCO_3$, 2.46 g sodium acetate, 8.0 g sodium fumarate, 10 mL Wolfe's vitamin solution and 10 mL modified Wolfe's minerals. The medium in sealed bottles was flushed with $N_2$-$CO_2$ (80:20) to remove oxygen before autoclaving.

## Analytical methods

Cell growth was monitored by measuring the optical density at 730 nm ($OD_{730}$) for cyanobacteria and at 600 nm ($OD_{600}$) for $E.\ coli$, $S.\ oneidensis$ and $G.\ sulfurreducens$ using a TU-1900 UV–VIS spectrophotometer (Persee, Beijing, China). The concentrations of lactate, acetate, fumarate and succinate were analyzed using an Agilent 1260 HPLC system (Agilent Technologies, CA, USA), equipped with a Bio-Rad HPX-87H column (Bio-Rad Laboratories, CA, USA) kept at 55 °C, with 5 mM $H_2SO_4$ as the mobile phase at a flow rate of $0.6\,mL\cdot min^{-1}$. The injection volume was 10 µL. Sucrose concentrations were measured using a Sucrose/D-Glucose Assay Kit (Megazyme, Bray, Ireland). Intracellular ROS was determined using a Fluorometric Intracellular ROS Kit (Genview, Florida, USA)[48]. A JPB-607A handheld Dissolved Oxygen Meter (Leici, Shanghai, China) was used to measure DO levels.

The drop-counting method was employed to determine the viability of $E.\ coli$ and $S.\ oneidensis$. Briefly, the precultured cells were adjusted to an $OD_{600}$ of 1.0, which was set as the undiluted culture (dilution factor 0), from which 10-fold serial dilutions were prepared with LB medium. Then, 10 µL of each dilution was dropped onto LB agar plates. The plates were incubated aerobically or anaerobically at 30 °C for colony formation.

## Solar energy allocation in engineered cyanobacteria

In the engineered cyanobacteria strain of Syn7942-FL130, the fixed solar energy was almost totally stored in sucrose and biomass. The energy stored in sucrose and biomass can be calculated according to the heat of combustion (also known as the heating value or the calorific value)[48]. The standard heat of combustion ($\Delta_c H^f$) of sucrose is 16.52 kJ/g[64]. The heating value of the biomass of cyanobacteria ($S.\ elongatus$) is 19.25 kJ/g[65]. The empirical factor for the conversion of $OD_{730}$ to dry cell weight (DCW) was determined to be 0.34 $gDCW\cdot L^{-1}$ per $OD_{730}$[66]. Therefore, the solar energy allocation into sucrose ($E_p$) was calculated using Eq. (1):

$$E_p = (16.52 \times M)/(16.52 \times M + 19.25 \times 0.34 \times OD_{730}) \times 100\% \quad (1)$$

where $M$ is the titer of sucrose ($g\cdot L^{-1}$). The value of solar energy allocation into biomass equals $1 - E_p$.

## Constructing synthetic communities attached to porous electrodes

RPPy was synthesized by the soft-template method[31]. Briefly, 300 mL $FeCl_3\cdot6H_2O$ aqueous solution (40 mM) was added slowly to 600 mL of methyl orange aqueous solution (5 mM) under magnetic stirring and maintained for 10 min. Then, 2.05 mL pyrrole was dropped into the above mixed solution. Lastly, the mixture was statically incubated at room temperature for 24 h. Carbonization was carried out in a high-temperature tube furnace by gradually heating the RPPy to 1600 °C at $2\,°C\cdot min^{-1}$ under a protective argon atmosphere and maintained for 3 h.

To construct the synthetic microbial communities, a dual-chamber electrochemical device separated by Nafion 117 proton exchange membranes (DuPont, USA) was used (Supplementary Fig. 19a, b). Carbon cloth (CeTech, Taiwan, China) was used as the cathode (3.0 cm × 3.0 cm). RPPy or carbon cloth (2.5 cm × 2.5 cm) was used as the anode. The cathodic electrolyte was composed of 100 mM $K_3[Fe(CN)_6]$ in 50 mM potassium phosphate buffer, pH 7.0. To inoculate different microorganisms, 150 mL of a culture of strain Syn7942-FL130 cultivated in MBG11-S medium for 15 d was transferred firstly into the anodic chamber, supplemented with 0.31 $g\cdot L^{-1}$ $NH_4Cl$, 0.5 $g\cdot L^{-1}$ tryptone and 0.25 $g\cdot L^{-1}$ yeast extract. Subsequently, the precultured cells of $E.\ coli$, $S.\ oneidensis$ and $G.\ sulfurreducens$ were centrifuged, washed and inoculated at an $OD_{600}$ of 0.005, 0.05 and 0.5, respectively. The above inoculation densities were determined by considering the differences in growth rate of three microbial species, and a relatively low inoculation density for $E.\ coli$ was designed to avoid the excessively competition for the anode surface. The devices were incubated in the dark at 30 °C without shaking. To measure the electrical current generated, the anode and the cathode were connected by an external resistor ($R$) of 2000 Ω or 510 Ω. The voltage ($U$) across the external resistor was recorded using a 2638 A Data Acquisition System (FLUKE, WA, USA). The current density ($I$) was calculated according to Ohm's law, using Eq. (2):

$$I = U/R \quad (2)$$

and normalized to the geometric area of the anode (6.25 cm²).

## Electrochemical characterizations

CV and EIS analyses were performed in a three-electrode configuration with an Ag/AgCl as reference electrode. CV analysis was conducted by using a CHI1030C potentiostat (CH Instruments, Shanghai, China) at a scan rate of $1\,mV\cdot s^{-1}$, associated with an electrochemical software chi1030c. EIS measurement was conducted using a CHI660E electrochemical workstation (CH Instruments, Shanghai, China) at open-circuit potential over an AC frequency range of 100 kHz to 1 mHz, with a sinusoidal perturbation of 5 mV. The Nyquist plots were fitted to an equivalent circuit model (Supplementary Fig. 20) using the ZSimpWin software.

LSV analysis was conducted with applied voltage ($U$) across the positive and negative terminals changed from the open-circuit potential (OCP, 880 mV) to 0 mV at a rate of $1\,mV\cdot s^{-1}$ controlled by the CHI1030C potentiostat, and the current ($I$) was recorded in real time. The output power ($P$) was derived via the Eq. (3). The current and power were normalized to the geometric area of the anode (6.25 cm²) to obtain the current density and power density, respectively.

$$P = U \times I \quad (3)$$

The constant-current discharge was conducted using a CT3001A LANHE battery test system (Wuhan LAND Electronics, China) associated with software LANDMon. The current across the positive and negative terminals was increased from 0.1 - 1.0 mA with a step of 0.1 mA. At each current value, the discharging was maintained for 1 h,

followed by an open-circuit state for 5 min. The voltage between the positive and negative terminals was recorded.

## Characterization of the anodic biofilms

Scanning electron microscopy (SEM) for bare electrodes and biofilm electrodes was carried out using an SU8010 field emission SEM instrument (Hitachi, Japan) at an accelerating voltage of 5 kV. The anodes after voltage recording ended were used for microbial diversity analysis. Total genomic DNA was extracted from the anode samples using a FastDNA™ SPIN Kit for Soil (MP Biomedicals, USA) following the manufacturer's instructions. The DNA content was determined using a NanoDrop 2000 Spectrophotometer (Thermo Fisher Scientific, USA). The V3-V4 variable region of the bacterial 16 S rRNA gene was amplified using the primers 338 F (5'-ACTCCTACGGGAGGCAGCAG-3') and 806 R (5'-GGACTACHVGGGTWTCTAAT-3'). Sequencing was performed using Illumina MiSeq technology (Illumina Inc., San Diego, CA, USA) at Majorbio Bio-Pharm Technology Co., Ltd. (Shanghai, China). The data were analyzed on the online platform of Majorbio Cloud Platform. Microbial diversity was calculated based on the relative abundance of OTUs (Operational Taxonomic Units).

## Fabrication of the miniaturized bionic ocean-battery

Agarose-gelatin hydrogel was prepared by adding 1.5% agarose (Sigma-Aldrich, V900510) and 1.5% gelatin (Amresco, 9764) into MBG11-S medium, and autoclaved at 115 °C. Hydrogel solidification occurred when the temperature decreased below 30-40 °C. To prepare the conductive hydrogel, 1.0% conductive material was added to the mixture of agarose and gelatin before autoclaving. The conductive materials used in this study included polypyrrole (PPy), polyaniline (PANI), carboxyl functionalized multi-walled carbon nanotubes (C-MWCNTs), amino functionalized multi-walled carbon nanotubes (N-MWCNTs), and conductive carbon black. PPy (catalog number: P871996) was purchased from Macklin, PANI (catalog number: B010625) was purchased from Energy Chemical, C-MWCNTs (catalog number: C139886) and N-MWCNTs (catalog number: C139874) were purchased from Aladdin, and conductive carbon black (catalog number: LG13-708) was purchased from Lige Science.

To construct miniaturized bionic ocean-battery, single-chamber electrochemical devices were used (Supplementary Fig. 19c, d). Carbon cloth W1S1009 (CeTech, Taiwan, China) containing 0.5 mg·cm$^{-2}$ of Pt catalyst was used as the air cathode (2.0 cm × 2.0 cm). In the chamber, the cells of *E. coli*, *S. oneidensis* and *G. sulfurreducens* were added to 30 mL of conductive hydrogel when the temperature decreased to ~40 °C, followed by mixing and solidification on ice. The OD$_{600}$ of *E. coli*, *S. oneidensis* and *G. sulfurreducens* was adjusted to 0.02, 0.5 and 0.5, respectively. Considering the growth of the encapsulated cells might be restricted inside a compact and anaerobic conductive hydrogel, the inoculation densities of *E. coli* and *S. oneidensis* were increased accordingly. Subsequently, 100 mL of MBG11-S medium was added to the chamber, supplemented with 50 mM NaHCO$_3$, 2 μM IPTG, 0.5 g·L$^{-1}$ tryptone and 0.25 g·L$^{-1}$ yeast extract, and then strain Syn7942-FL130 was inoculated with an OD$_{730}$ of 0.6. The anode (conductive hydrogel) and the air cathode were connected by an external resistor of 2000 Ω. The whole devices were placed in an illumination incubator (30 °C, 3% CO$_2$, 20 W·m$^{-2}$ white light-emitting diodes) and magnetically stirred at 250 rpm. The voltage across the external resistor was recorded using a 2638 A Data Acquisition System (FLUKE, WA, USA). The calculation of current densities was normalized to the geometric area of receiving the light (60 cm$^2$).

## The calculation of energy conversion efficiency

The coulombic efficiency ($\eta_F$) was defined as the ratio of the actual amount of Coulombs generated ($Q$) to the maximum amount of Coulombs released by the oxidation of substrates ($Q_{max}$). Thus, the coulombic efficiency of the synthetic microbial communities was calculated using Eq. (4):

$$\eta_F = Q/Q_{max} = \left( \int_0^T i \cdot \mathrm{d}t \right) / (n \cdot m \cdot F) \tag{4}$$

where $n$ is the number of electrons released when one molecule of sucrose is oxidized to acetate ($n = 16$) or $CO_2$ ($n = 48$), $m$ (mol) is the molar amount of sucrose consumed, and $F$ is the Faraday constant (96485 C·mol$^{-1}$ of electrons). The value of $\int_0^T i \cdot \mathrm{d}t$ is the integral area of $i$–$t$ curves.

To elucidate the overall energy balance of the miniaturized bionic ocean-battery, the energy flux and energy efficiency of three sub-processes, including light to sucrose, sucrose to lactate, and lactate to electricity, were calculated respectively. The light energy input was 20 W·m$^{-2}$, and the energy fluxes of sucrose and lactate were calculated based on the Gibbs free energy of the oxidation of sucrose or lactate into $CO_2$. The values of energy flux were normalized to the geometrical area of receiving the light (60 cm$^2$).

## Calculating the power density of global marine ecosystems

The average power density of global marine ecosystems ($P_g$) was calculated according to the combustion heat of net primary production (NPP) in the oceans. Annual NPP in the oceans was estimated to correspond to 48.5 Pg of carbon per year[3]. The elemental composition of carbon in organic matter was estimated to be 50%, and the heating value of organic matter was estimated to be 20 kJ/g in a recent study[65]. The area of global oceans is ~3.625 × 10$^8$ square kilometers[67]. The calculation of $P_g$ refers to Eq. (5):

$$P_g = \frac{\left(2 \times 48.5 \times 10^{15} \times 20000\right)}{\left(365 \times 86400 \times 3.625 \times 10^{14}\right)} = 0.169 \ \mathrm{W \cdot m^{-2}} \tag{5}$$

## Reporting summary

Further information on research design is available in the Nature Research Reporting Summary linked to this article.

# Data availability

Data supporting the findings of this work are available within the paper and its Supplementary Information files. A reporting summary for this Article is available as a Supplementary Information file. Source data are provided with this paper.

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

## Acknowledgements

This work was supported by the National Natural Science Foundation of China (22072180, Y.Z.), the DNL Cooperation Fund, CAS (DNL202014, Y.Z.), the Strategic Priority Research Program of the Chinese Academy of Sciences (XDPB18, Y.L.), the Key Research Program of the Chinese Academy of Sciences (ZDRW-ZS-2016-3, Y.L.), and the Projects funded by China Postdoctoral Science Foundation (BX20220333, 2022M710161, H.Z.). We thank H. Song from Tianjin University for providing the strain *E. coli* WM3064; H.C. Gao from Zhejiang University for providing the strain *S. oneidensis-ΔnapA*.

## Author contributions

H.Z., Y.Z. and Y.L. conceived the project. H.Z. and L.X. developed the conductive hydrogels. H.Z., G.L., T.Z., X.L. and X.Z. constructed the engineered strains. Z.K. and Z.Z. prepared the RPPy electrode. C.L. conducted SEM characterizations. H.Z. performed all other experiments. H.Z., Y.Z. and Y.L. analyzed the experimental data. H.Z. and Y.L. wrote the manuscript. All authors critically revised the article and approved the final manuscript.

## Competing interests

A patent has been filed by the Institute of Microbiology, CAS, which covers the fabrication of conductive hydrogel-based bio-solar cell (application No. 2021113830301.1; inventors: Y.L., H.Z., Y.Z. and L.X.). The other authors declare no competing interests.
