## [Peer Review File · Nature Communications]

A miniaturized bionic ocean-battery mimicking the structure of marine microbial ecosystemsReviewers' Comments:

Reviewer #1:

Remarks to the Author:

The article describes an interesting artificial consortia inspired on oceanic ones and proposes it as a new type of ocean-battery. However, There are major concerns on the feasibility of this idea, on the very notion/writing of its ecological concepts, and on some of the figures displaying the electric properties of the bio-solar cells. A detailed list of comments and issues are listed below.

SUMMARY

" Marine ecosystems can be viewed as a huge rechargeable ocean-battery powered by solar energy " that power...what, exactly?

"bio-solar cells". Please define this here.

"power density of $1.7 \text{ W}\cdot\text{m}^{-2}$ from light/maximum power of $860 \mu\text{W}$ ". How does this compare with a regular MFC? And with other bio-photoelectrochemical cells, such as the one reported here:

<https://pubmed.ncbi.nlm.nih.gov/34864244/>

INTRODUCTION

L 38-40. I guess organic matter is mostly used in the food chain and recycled as source of nutrients por phototrophs rather than directly sinking. Please clarify.

L 41. "For complete remineralization of organic matter, two types of heterotrophic microorganisms work together in the sediments". Do they really "work together to achieve a common goal"? I would use with care such engineering metaphors.

L 54. Here too, the writing style suggests a purpose in the process: "the photosynthetic microorganisms in the ocean's surface area are irradiated by solar energy..." I suggest, instead "use solar energy...". The rest of the paragraph, until line 62, albeit suggestive and interesting, also flows between the battery metaphor and the notion of purpose, which I personally do not find suitable for the description of an ecological system.

L 65 " the upper sediment layers below the seafloor". Since upper-below can be confusing, consider removing "below" and use "of" instead.

L 70. I think I understand what the authors mean here (an intricate, often indirect transfer of electrons among microbial species in a network-like way), but "non linear" is a misleading expression.

L 75. "Inspired by the sustainable and rechargeable ocean-battery, we aimed to develop a bio-solar cell ". In line with my previous comments, I would, at least, add "concept", "hypothesis" or "metaphor" after "ocean-battery".

L 83. "Following this basic structure, we developed a synthetic microbial community composed of three ecological niches". I am not sure that those metabolic profiles are exactly "ecological niches", since I guess the concept of niche refers to the environment, rather than to organisms/metabolic profiles.

L84-end. What is the goal of such bio-cell?

RESULTS

The choice of a non-marine organism such as E. coli as member of an artificial consortium mimicking the "ocean battery" concept is odd. Please clarify.

The spatial distribution of the bacteria is showed in figure 2 as "ideal models". Scanning Electron Microscopy or another technique are preferable to find out the actual distribution. Fig 2 G. Signal blunt changes after 6 and 12 days are odd and should be explained with detail. Fig 2 E also exhibits a strange behavior at time 6 days and an odd change in the trend of C+S2 is seen at time 6 days in Fig 2F. All these graphs do not seem to follow the expected trends of a continuous measuring but, instead, seem the result of pasting together different (weekly?) experiments. Please clarify.

Figure 3. I am not familiar with this kind of representations, but I assume that the figure displays changes in Voltage as Intensity increases. How is it possible (a,b) that the curves sometimes (C+E2+S1+G) bend backwards!?

The data on the relative proportion of primary producers (cyanobacteria) vs. heterotrophic strains (the other three species), as shown in Figure 4, yield to the conclusion that primary producers (typically

the large bases of the ecological model) are really minoritarian on the anodic biofilms. If I am not wrong, this is solved by cultivating cyanobacteria in the top part of the system. What about the oxygen produced by cyanobacteria on the anodic biofilms? Wouldn't that disturb the anaerobic microorganisms there?

DISCUSSION

Where dark controls used? That would be useful to remove the Microbial Fuel Cell (MFC)-like power production in the system. In other words, part of the power produced may not be linked to the light-to-power expected equation, but to microbial oxidation of reduced compounds from the starting biomass, as it is the case in MFCs.

"In this miniaturized ocean-battery, the four species affiliated to three-trophic 284 levels cooperated with each other to accomplish photoelectric conversion." Please consider my comments above on the use of such anthropocentric metaphors.

"Among the three ecological niches, the primary degraders serve as a connecting link between primary producers and ultimate consumers". Again, I think there is a confusion in this article between i) natural ecological niche; ii) artificial ecological niche and; iii) microorganisms (either natural or engineered) actually "filling" such niches.

Finally, all the discussion on the bio-solar cells based on synthetic microbial communities such as the one described in the article remains ambiguous. What for? At what scale could the system be developed? Where and how would be the required facilities? And, importantly, how is it possible to implement an ocean-like system partially based on *E. coli* rather than native, halophilic microorganisms? How would the engineered, artificial consortium evolve in time with/without selection pressures? Which can of facilities could be powered with such a technology?

Reviewer #2:

Remarks to the Author:

In this manuscript, the authors introduce the concept of "ocean-battery" visualizing the energy conversion processes in the marine environment as a bio-solar cell. The authors aim to reconstruct the "ocean-battery" in a simplified manner using bioengineering approaches. They create a compact, bio-inspired battery using primary producers (engineered *Synechococcus elongatus*), primary degrader (engineered *E. coli*) and ultimate consumers (*S. oneidensis* and *G. sulfurreducens*). Overall, the study is well conceptualized and the data are presented in a logical manner. I have the following comments for the authors to address:

1. *Geobacter sulfurreducens* and *Escherichia coli* are not representative bacteria in the marine environment. The term "miniaturized ocean-battery" used in the title and main text is misleading. The battery designed and demonstrated in this study seems to be inspired by microbial ecology in the ocean.
2. Figure 1 shows metabolic interaction in the marine ecosystem and the synthetic microbial communities. For the synthetic communities, it is challenging to optimize the flow of metabolites by constructing the microbial cocktails with carefully designed relative abundance of each organism. How was the ratios used in this study determined, for example, Line 546: "The 546 OD600 of *E. coli*, *S. oneidensis* and *G. sulfurreducens* was adjusted to 0.02, 0.75 and 0.75, respectively"? Discussion on this aspect is needed.
3. The authors claimed that their system "generated a maximum power density of 1.7 W·m⁻² from light, which is tenfold higher than the estimate for marine ecosystems". Is the power density an average value of several reactors or just the result of one reactor? Statistical analyses are missing for many key results reported in this paper. For example, error bars are missing in Figure 4. In addition, the energy and coulomb efficiencies should also be reported.
4. Key electrochemical analyses are lacking: To probe into the mechanism of the constructed system,

CV and EIS analyses should be included, rather than simply comparing resistances from polarization curves.

5. The authors stated that the battery is "self-sustaining and self-reproducing" and can even be used on Mars. However, the basic cycle stability tests are lacking in the manuscript. The systems collapsed only after 20 to 30 days of operation. In addition, the reason for these collapses and how to maintain a long-term stability should be discussed in the manuscript.

6. What was the power source utilized for light energy? Does the light used here mimic the sunlight? Impacts of diurnal cycles?

7. Line 375: Why was the M1-93 and rrB promoter chosen for the strain engineering?

8. Some of the references used for the methods are not properly mentioned. For examples, lines 560, 469, 470 etc.

9. Figure S9: It is hard for the audience to visualize what the authors are pointing out. The electrographs look very similar to me and the presence of biofilms cannot be verified using these images.

10. Figure S11: Is the transfer of lactate and sucrose a limiting factor in the design of the hydrogel battery?

11. The introduction might be slightly misleading. While organic material transfer to heterotrophic organism is mentioned, it is unclear what are the organisms. For example, primary producers (carbon dioxide fixers by phototrophs and chemolithoautotrophs) are commonly predated by other planktons (metazoans like fish larvae and crustaceans) that drives the biological pump that ultimately stores carbon in the deeper ocean (current introduction seems to imply most of biogeochemical cycles only involve microbes). Hence, I think this portion should be clarified, and then focusing "ocean-battery" that is driven by microbial loop.

12. Line 58: Might want to take note of mentioning of methane production through decomposition and methanogenesis, especially in the deep ocean.

13. Line 88: Incomplete sentence.

Reviewer #3:

Remarks to the Author:

The present study describes the construction and operation of a novel bio-solar cell consisting of a four-species microbial community. This novel system mimics the basic ecological structure of marine ecosystems. The authors claimed that a three-dimensional electrode (carbonized conductive polymer polypyrrole) colonised with the four-species microbial community generated a maximum power density of $1.7 \text{ W}\cdot\text{m}^{-2}$ from light.

Furthermore, the authors also described the construction and operation of a miniaturized ocean-battery, making the claim that this battery directly converts light into electricity with a maximum power of $860 \mu\text{W}$.

The concept of "Marine ecosystems as rechargeable ocean-battery" as well as the term of "ocean-battery" are very interesting, I am complimenting with the authors for introducing these new ideas and having such a clear and well-made graphical materials (e.g., figure 1, 6 and 7).

In my view the present study deserves to be published in Nature Communication if the claim of "power output from light" is proven. This can be done by:

- 1) running the system operated with C+S2+S1+G in the dark and doing a comparison with an equal system operated under light condition;
- 2) running the system operated with C+S2+S1+G in the light after having consumed all the metabolite(s) introduced with the media used for growing the microorganisms (e.g., LB medium is rich in organics).
- 3) running the system operated with S2+S1+G (without cyanobacteria) under light condition and doing a comparison with an equal system operated with C+S2+S1+G (with cyanobacteria) under light condition;
- 4) calculating the power density by using the geometrical area receiving the light, not the geometric area of the anode (more details in my comment 6).

Additional comments

5) Page 7 and 8, line 129 - 134

The authors have defined the carbonizing conductive polymer polypyrrole as a 3D electrode. In comparison they have labelled the carbon cloth as "planar" (therefore not 3D).

However, by looking at the Supplementary Fig. 9, both electrodes seemed to me very similar in terms of "porosity"/morphology. Could the authors provide any analytical quantification of the electrodes' porosity/morphology? If not, both electrodes have to be named either 3D or planar.

Also, by looking at the Supplementary Fig. 4, the electrical output (as Coulomb) of RPPy looks to me very similar to the electrical output of CC. Therefore, unless the authors can prove a statistical difference, those two electrodes displayed an equivalent electrical output. Could the authors comment on this?

6) Abstract and main text

In the abstract, and various parts of the main text, the authors have claimed "the synthetic community inside a three-dimensional electrode generated a maximum power density of 1.7 W·m⁻² from light".

According to the Method (line 499-501), this was calculated according to Ohm's law and normalized to the geometric area of the anode (6.25 cm²).

This is in my view not correct. As the authors claim "from light", the normalization has to be based on the geometrical area receiving the light, not the geometric area of the anode.

7) More details of the electrochemical apparatus are required

Please provide in the main text more details for the electrochemical apparatus used for doing the experimental work. For example, actual dimensions and geometry. I suggest to provide a schematic as well as an actual photo of the apparatus (those could be displayed in the supplementary material).

8) The overall energy balance.

As the light intensity, amount of metabolites and electrical output are known, please provide the overall energy balance of the presented system.

Please start from light (W/m²) going into sucrose (mmol/m²) and then acetate/acetate (mmol/m²) for then finishing with current (mW/m²).

9) Figure 6d

In the figure 6d, the authors presented the performance comparison of this study with previously demonstrated bio-solar cells, showing how their new system exceeded all the previously tested. This comparison is not very informative as the data could come from devices with a very different size.

Please change this as shown in the Supplementary figure 7 (power density)

10) Was the light ON?

The experimental data used to generate the data shown in figure 2d-g, 5b and 6b were obtained with light ON or OFF? Please also indicate the light intensity and quality.

Also, please provide results (e.g., chronoamperometry) showing few dark/light cycle.

Dr. Paolo Bombelli.

REVIEWER COMMENTS

Reviewer #1 (Remarks to the Author):

The article describes an interesting artificial consortia inspired on oceanic ones and proposes it as a new type of ocean-battery. However, there are major concerns on the feasibility of this idea, on the very notion/writing of its ecological concepts, and on some of the figures displaying the electric properties of the bio-solar cells. A detailed list of comments and issues are listed below.

SUMMARY

“Marine ecosystems can be viewed as a huge rechargeable ocean-battery powered by solar energy” that power...what, exactly?

Response: Thank you very much for your comment. Here the “power” refers to “charge”. We deem the process that marine ecosystems receive solar energy is a charging process, where the photosynthetic organisms on the surface of the ocean absorb the photons from sunlight and store the energy into organic matter. To clarify this, “powered” was modified as “charged”. Please refer to Line 23.

“bio-solar cell”. Please define this here.

Response: Thank you very much for your comment. The term “bio-solar cell” was defined as “an electrochemical system that converts light into electricity using microorganisms” in the Abstract (Line 24-25).

“power density of $1.7 \text{ W}\cdot\text{m}^{-2}$ from light/maximum power of $860 \mu\text{W}$ ”. How does this compare with a regular MFC? And with other bio-photoelectrochemical cells, such as the one reported here: <https://pubmed.ncbi.nlm.nih.gov/34864244/>

Response: Thank you very much for your comment. The power densities of the MFCs reported to date were generally between 0.2 and $2 \text{ W}\cdot\text{m}^{-2}$. A latest study (DOI: 10.1126/science.abf3427) reported a power density of $6.6 \text{ W}\cdot\text{m}^{-2}$, which was the highest in MFCs ever reported. Thus, the power density of $1.7 \text{ W}\cdot\text{m}^{-2}$ reported in this study was comparable to a regular MFC. A comparison between our study and MFCs was added in the Discussion (Line 334-335).

The bio-photoelectrochemical cell reported recently by Shlosberg *et al.* (DOI: 10.1016/j.bios.2021.113824) uses photosynthetic macroalgae for bioelectricity generation and produced a relatively high photocurrent with a bias added. The electrochemical device installed in an *Ulva* cultivation tank produced a maximal current of ~0.5 mA at an open circuit potential of 0.38 V. The calculated power output of a single device was 190 μ W, which was 2-fold lower than the power output (380 μ W) of the miniaturized bionic ocean-battery presented in this study. This article has been cited in the Supplementary Data 1. A brief comparison between our study and Shlosberg *et al.* was added in the Discussion (Line 340-342).

INTRODUCTION

L 38-40. I guess organic matter is mostly used in the food chain and recycled as source of nutrients for phototrophs rather than directly sinking. Please clarify.

Response: Thank you very much for your comment. Indeed, a large fraction of organic matter produced by primary producers was taken up by heterotrophic plankton (including heterotrophic microorganisms, zooplankton, etc.) lived in oxic zones of water column, only a part of organic matter was deposited into the sediments. We amended the description as “Organic matter can be consumed by heterotrophic plankton lived in the water column, or deposited into the marine sediments through sinking and burial” in Line 45-46.

L 41. “For complete remineralization of organic matter, two types of heterotrophic microorganisms work together in the sediments”. Do they really “work together to achieve a common goal”? I would use with care such engineering metaphors.

Response: Thank you very much for your suggestion. To avoid confusion, we have modified this description as “Marine sediments is a large anaerobic bioreactor where organic matter is slowly degraded and fully oxidized by two types of heterotrophic microorganisms, eventually achieving complete remineralization” in Line 46-48.

L 54. Here too, the writing style suggests a purpose in the process: “the photosynthetic microorganisms in the ocean’s surface area are irradiated by solar energy...” I suggest, instead “use solar energy...”. The rest of the paragraph, until line 62, albeit suggestive and interesting, also flows between the battery metaphor and the notion of purpose,

which I personally do not find suitable for the description of an ecological system.

Response: Thank you very much for your suggestion. We have modified the description as “the photosynthetic microorganisms in the ocean’s surface use solar energy to fix carbon dioxide into organic matter.” Please refer to Line 62-63.

The aim of this paragraph was to define the ocean-battery concept so as the readers can easily follow the logic. We agree with the reviewer not to overuse battery metaphor, so we deleted “The electron flow among heterotrophic microorganisms resembles the electric current in the external circuit of a battery.”

L 65. “the upper sediment layers below the seafloor”. Since upper-below can be confusing, consider removing “below” and use “of” instead.

Response: Thank you, modified as suggested. Please refer to Line 71.

L 70. I think I understand what the authors mean here (an intricate, often indirect transfer of electrons among microbial species in a network-like way), but “non linear” is a misleading expression.

Response: Thank you very much for your suggestion. The “linear” and “non-linear” are actually mathematical terms, thus may not be appropriate to be used. Following your advice, this sentence was modified as “Moreover, the ocean-battery is a highly intricate system where electrons are transferred among microbial species in a network-like way due to its vast microbial diversity and incredibly complex interspecies interactions” (Line 76-78). Moreover, the adjectives “linear” and “non-linear” throughout the manuscript were modified accordingly.

L 75. “Inspired by the sustainable and rechargeable ocean-battery, we aimed to develop a bio-solar cell”. In line with my previous comments, I would, at least, add “concept”, “hypothesis” or “metaphor” after “ocean-battery”.

Response: Thank you very much for your suggestion. We have modified the description as “ocean-battery concept”. Please refer to Line 82.

L 83. “Following this basic structure, we developed a synthetic microbial community

composed of three ecological niches”. I am not sure that those metabolic profiles are exactly “ecological niches”, since I guess the concept of niche refers to the environment, rather than to organisms/metabolic profiles.

Response: Thank you very much for your comment. It’s true that a specific organism cannot equal to an ecological niche. Thus, we modified the description as “Following this basic structure, we design a synthetic microbial community composed of specific microorganisms affiliated to three ecological niches.” in Line 90-92.

L 84-end. What is the goal of such bio-cell?

Response: Thank you very much for your question. This sentence has been simplified as “Using conductive hydrogel as a sediment-like matrix, we fabricate a miniaturized bionic ocean-battery with marine microbial ecological structure, which can stably convert light into electricity”, with the goal of this study specified. Please refer to Line 94-94.

RESULTS

The choice of a non-marine organism such as *E. coli* as member of an artificial consortium mimicking the “ocean battery” concept is odd. Please clarify.

Response: Thank you very much for your comment.

In this work, *E. coli* was chosen as a member of the artificial consortium because of the following considerations. Firstly, *E. coli* is a well-known fermentative bacterium, which can convert sugars into various fermentation products including lactate under anaerobic conditions. Secondly, although the primary habitat of *E. coli* is thought to be the lower intestine of warm-blooded animals, recent studies have reported *E. coli* persisted autochthonously in environment matrices such as sediments, sands and soils without any known association with fecal contamination (Ref: 10.1021/es0623156; 10.1111/jam.13468; 10.1093/femsec/fix187; 10.1038/nrmicro1158). This means *E. coli* is also a member of microbial communities in the natural environments. Thirdly, *E. coli* is one of the most frequently employed host microorganisms in biotechnology due to its advantages of fast growth, easiness for cultivation, and well-established genetic engineering tools, which allows us to easily engineer it according to our experimental design.

Therefore, we deem *E. coli* is a suitable primary degrader (fermentative microorganism) to be used in the artificial microbial community. The relevant description was added in Line 105-107.

The spatial distribution of the bacteria is showed in figure 2 as “ideal models”. Scanning Electron Microscopy or another technique are preferable to find out the actual distribution.

Response: Thank you very much for your comment. We have tried to use Confocal Laser Scanning Microscopy (CLSM) to observe the actual distribution of the bacteria, but failed due to the interference of the black electrode materials. We added a sentence in the revised manuscript (Line 156-157), explaining our effort and the possible reason for the failure. Understanding the spatial distribution of all bacteria graphically would be very interesting, however it is also very challenging and we will explore this in future studies.

Fig 2 G. Signal blunt changes after 6 and 12 days are odd and should be explained with detail. Fig 2 E also exhibits a strange behavior at time 6 days and an odd change in the trend of C+S2 is seen at time 6 days in Fig 2F. All these graphs do not seem to follow the expected trends of a continuous measuring but, instead, seem the result of pasting together different (weekly?) experiments. Please clarify.

Response: Thank you very much for your comment. The signal fluctuations in Fig 2d-g were caused by the linear sweep voltammetry (LSV) scanning. In Fig 2d and 2f, LSV scanning was conducted at day 5. In Fig 2e and 2g, LSV scanning was conducted at day 5.5 and day 12. The great change of electrical current during LSV scanning would stimulate the electroactive biofilms, thus may lead to the change of pseudo-steady-state, especially for the immature biofilms. The greatest signal blunt change shown in Fig. 2g indicated that the biofilm formed on carbon cloth was possibly the least stable, thus is most sensitive to fluctuations introduced by potential change.

The relevant detail was clarified in the figure legend (Line 853-854). We also supplemented more interpretations for this phenomenon in the main text (Line 151-154).

Figure 3. I am not familiar with this kind of representations, but I assume that the figure displays changes in Voltage as Intensity increases. How is it possible (a,b) that the curves sometimes (C+E2+S1+G) bend backwards!?

Response: Thank you very much for your question. The appearance of bending inwards in polarization/power curves is a common phenomenon for a bioelectrochemical system, which is also termed as “power overshoot” (Ref: 10.1016/j.biortech.2009.12.108). According to the hypothesis of some previous studies, the demand for electrons of the external circuit at high current condition may exceed the electron generating rate of microorganisms. This may be one of the reasons why current decreases when it is expected to increase. The relevant interpretation for this phenomenon was added in Line 205-207.

The data on the relative proportion of primary producers (cyanobacteria) vs. heterotrophic strains (the other three species), as shown in Figure 4, yield to the conclusion that primary producers (typically the large bases of the ecological model) are really minoritarian on the anodic biofilms. If I am not wrong, this is solved by cultivating cyanobacteria in the top part of the system. What about the oxygen produced by cyanobacteria on the anodic biofilms? Wouldn't that disturb the anaerobic microorganisms there?

Response: Thank you very much for your question. Since the electricity was generated solely by *Shewanella* and *Geobacter*, thus the minority attachment of cyanobacteria on the anodic biofilms is actually advantageous, which avoided the competition for the anode surface.

To avoid the disturbance of oxygen produced by cyanobacteria to the anodic biofilms, the synthetic microbial communities using RPPy or CC as anode were constructed using a mode of temporal separation organization, which allowed the charging process (photosynthetic sucrose production) and discharging process (anaerobic sucrose oxidation) implemented sequentially under light and dark conditions, respectively. The relevant experimental details can be found in the Methods (Line 557-565), and we also clarified this in the Results (Line 142-147).

To address the oxygen contradiction, the conductive hydrogel was developed to isolate the oxygen produced by upper cyanobacteria in the following experiments. Thus, we eventually constructed an all-in-one system without the requirement of temporal separation mode (Fig. 5a and Fig. 6a).

DISCUSSION

Where dark controls used? That would be useful to remove the Microbial Fuel Cell (MFC)-like power production in the system. In other words, part of the power produced may not be linked to the light-to-power expected equation, but to microbial oxidation of reduced compounds from the starting biomass, as it is the case in MFCs.

Response: Thank you very much for your great comment. Similar questions were raised by Reviewer 3. To address this, we supplemented additional experiments to further demonstrate that the power output was from light.

Specifically, the control experiments (i.e. dark control, cyanobacteria-free control and *E. coli*-free control) were performed in the conductive hydrogel-based systems. As shown in Fig. 6b, all the systems (including experimental group) generated an electrical current peak during the early two days, which could be ascribed to the residual metabolites or a bit of organic matter introduced by the medium. Nevertheless, all the electrical currents of three control groups decreased after two days, whereas it increased to a high level for the experimental group due to the photosynthetic sucrose production. In addition, the photo-response of electrical current was observed in the light-dark cycles (Supplementary Fig. 17). All these results proved that the power output was from light. The relevant descriptions can be found in Line 291-306.

“In this miniaturized ocean-battery, the four species affiliated to three-trophic levels cooperated with each other to accomplish photoelectric conversion.” Please consider my comments above on the use of such anthropocentric metaphors.

“Among the three ecological niches, the primary degraders serve as a connecting link between primary producers and ultimate consumers”. Again, I think there is a confusion in this article between i) natural ecological niche; ii) artificial ecological niche and; iii) microorganisms (either natural or engineered) actually “filling” such niches.

Response: Thank you very much for your comments. We have removed the ocean-battery metaphor from the Introduction.

We are aware of the confusion between microorganisms and ecological niches. The correct should be that a specific microorganism fills or occupies the corresponding ecological niche, rather than equals to a specific ecological niche.

To avoid this confusion, the above descriptions were revised as following: “In this miniaturized bionic ocean-battery, the four microbial species with different functions worked in a cascade way to accomplish photoelectric conversion. Among these microorganisms, the primary degraders serve as a connecting link between primary producers and ultimate consumers.” Please see Line 347-350.

Finally, all the discussion on the bio-solar cells based on synthetic microbial communities such as the one described in the article remains ambiguous. What for? At what scale could the system be developed? Where and how would be the required facilities? And, importantly, how is it possible to implement an ocean-like system partially based on *E. coli* rather than native, halophilic microorganisms? How would the engineered, artificial consortium evolve in time with/without selection pressures? Which can of facilities could be powered with such a technology?

Response: Thank you very much for your inspiring questions. Indeed, addressing these questions will help the readers better understand the potential applications of this study.

Powering low-power electronic facilities using bio-solar cell has been demonstrated in several previous studies. Considering the potential applications of the bio-solar cell developed in this study, it is firstly hoped to serve as an alternative electrical energy source for ultralow-power electronic facilities, such as environmental sensors of Internet of Things (IoT). Since the IoT sensors consume power in the range of μW to mW , a single miniaturized bionic ocean-battery generating hundreds of μW is sufficient to support these small facilities.

Moreover, similar to the widely used solar cell, series-parallel stacking of bio-solar cells is one of the possible solutions to obtain the desired voltage and current output to power higher-power facilities, such as mobile phone. Installing bio-solar arrays on the building roofs or floating on the sea is a potential scene for its large-scale application. Furthermore, the large-scale fabrication of bio-solar cell is feasible by using 3D printing technology, especially for conductive hydrogel-based bio-solar cell.

As for the stability of the system, we have shown that the system developed in this study could stably run for more than 30 days. Based on these data, we deem that the encapsulated microorganisms would be able to work for a long period under the protection of conductive hydrogel, given that the activity of cyanobacteria can be

maintained. In addition, all the engineered microorganisms used in this study were genetically stable, and they were also adaptive to the high-salinity environments, because the salinity of medium used in this study was almost the same as that in the seawater. However, the practical output and robustness of such a bio-solar cell in the natural environments remains to be investigated.

We have added these discussions in Line 410-422.

Reviewer #2 (Remarks to the Author):

In this manuscript, the authors introduce the concept of “ocean-battery” visualizing the energy conversion processes in the marine environment as a bio-solar cell. The authors aim to reconstruct the “ocean-battery” in a simplified manner using bioengineering approaches. They create a compact, bio-inspired battery using primary producers (engineered *Synechococcus elongatus*), primary degrader (engineered *E. coli*) and ultimate consumers (*S. oneidensis* and *G. sulfurreducens*). Overall, the study is well conceptualized and the data are presented in a logical manner. I have the following comments for the authors to address:

1. *Geobacter sulfurreducens* and *Escherichia coli* are not representative bacteria in the marine environment. The term “miniaturized ocean-battery” used in the title and main text is misleading. The battery designed and demonstrated in this study seems to be inspired by microbial ecology in the ocean.

Response: Thank you very much for your comments. Yes, the ocean-battery concept was indeed inspired by the microbial ecology in the ocean. We have modified the “marine ecosystems” into “marine microbial ecosystems” throughout the manuscript. As you would agree, if the marine microbial ecosystems can be considered as an ocean-battery, it must comprise primary producer, primary degrader, and ultimate consumer. Generally speaking, microbes playing the functions of primary producer, primary degrader, and ultimate consumer can be chosen to construct the bio-solar cell following the ocean-battery concept.

In our study, we chose photosynthetic cyanobacterium (*Synechococcus*) as primary producer, fermentative bacterium (*Escherichia*) as primary degrader, metal reducing

bacteria (*Geobacter* and *Shewanella*) as ultimate consumers. The genus *Geobacter* and *Shewanella* are important metal reducing microorganisms in the marine sediments (Ref: 10.1038/nrmicro3347; 10.1038/nrmicro1490; 10.1111/1462-2920.14260; 10.1002/celc.201600079; 10.1146/annurev.genet.38.072902.091138), while *Synechococcus* and *Escherichia* are found in aquatic ecosystems or aquatic sediments (Ref: 10.1021/es0623156; 10.1111/jam.13468; 10.1093/femsec/fix187, also refer to the response to Reviewer 1). From the viewpoint of synthetic biology, we deem the microbes chosen to construct the ocean-battery should not be limited from the marine microbial ecosystems. Most importantly, the microbes chosen should play the desired ecological functions.

As for the words “miniaturized ocean-battery” used in the title and main text, a miniaturized ocean-battery refers to a miniature version of the ocean-battery. We first conceptualized ocean-battery and stated the aim of this study was to simplify the ocean-battery according to the basic microbial ecological structure of the marine ecosystems. As shown in Fig. 7, we were able to fabricate a miniature version of the ocean-battery by mimicking the microbial ecological structure, so we chose “miniaturized ocean-battery” to describe this bio-solar cell. We appreciate your comments and thought about other names including “compact ocean-battery”, “bio-spired ocean-battery”, or “microbial ocean-battery”, but it seems that these terms could not reflect the efforts of simplifying the ocean-battery.

Considering your comments and the fact that not all microorganisms in the bio-solar cell are from marine environments, we chose to use “miniaturized bionic ocean-battery” in the revised manuscript according to the context of this research. We hope you would agree with our consideration, but we are happy to modify this term if you could offer us a better concise definition, thank you.

2. Figure 1 shows metabolic interaction in the marine ecosystem and the synthetic microbial communities. For the synthetic communities, it is challenging to optimize the flow of metabolites by constructing the microbial cocktails with carefully designed relative abundance of each organism. How was the ratios used in this study determined, for example, Line 546: "The 546 OD600 of *E. coli*, *S. oneidensis* and *G. sulfurreducens* was adjusted to 0.02, 0.75 and 0.75, respectively"? Discussion on this aspect is needed.

Response: Thank you very much for your comments. The inoculation ratio of *E. coli* (E),

S. oneidensis (S) and *G. sulfurreducens* (G) in this study was determined by considering their physiological properties.

In the synthetic communities attached to porous electrodes, the inoculation optical density of E, S and G was designed as 0.005, 0.05 and 0.5, respectively. There are three considerations for that: (1) among three species, the growth rate of *E. coli* is faster than *S. oneidensis*, while *G. sulfurreducens* is the slowest. The faster the growth speed of the strain, the less the amount of cells inoculated. (2) *E. coli* and *S. oneidensis* can grow in co-culture medium because the medium contains a small amount of LB, whereas *G. sulfurreducens* cannot grow in co-culture medium, so we need to inoculate more *G. sulfurreducens* cells. (3) *E. coli* is not generating electricity, so large inoculation of *E. coli* may compete for the anode surface. To avoid this potential problem, the inoculation density of *E. coli* should be controlled at a low level. Based on the above considerations, we designed the inoculation density with one order of magnitude difference among E, S and G.

In the synthetic communities encapsulated by conductive hydrogel, the inoculation density of E, S and G was set as 0.02, 0.5 and 0.5, respectively. In this case, we thought the growth of the encapsulated cells might be restricted in such a compact and anaerobic space, and the cells are nearly non-motile. Thus, the inoculation density of *S. oneidensis* was increased from 0.05 to 0.5, and that of *E. coli* increased from 0.005 to 0.02. For *E. coli*, its inoculation density still should be controlled at a relatively low level to avoid the excessively competition for hydrogel pores, which serve as anode. According to our experience, the inoculation density of 0.02 was sufficient for *E. coli* to ferment sucrose to lactate.

The considerations for determining the inoculation density of three strains were supplemented in Methods (Line 562-565 and Line 622-624).

3. The authors claimed that their system “generated a maximum power density of 1.7 W·m⁻² from light, which is tenfold higher than the estimate for marine ecosystems”. Is the power density an average value of several reactors or just the result of one reactor? Statistical analyses are missing for many key results reported in this paper. For example, error bars are missing in Figure 4. In addition, the energy and coulomb efficiencies should also be reported.

Response: Thank you very much for your comments. The experiments in this study were performed in triplicates. But in some cases, such as Fig. 2, the inclusion of error bar will influence its clarity, thus the representative curves from one reactor were presented. The error bars in Figure 4 were added in the revised version. The coulomb efficiencies were calculated and presented in Supplementary Fig. 5b. The energy efficiencies of conductive hydrogel-based bio-solar cell was calculated and presented in Fig. 6d. The relevant descriptions were added in the main text. Please refer to Line 166-169 and Line 313-317.

4. Key electrochemical analyses are lacking: To probe into the mechanism of the constructed system, CV and EIS analyses should be included, rather than simply comparing resistances from polarization curves.

Response: Thank you very much for your comments. We supplemented the experiments of cyclic voltammetry (CV) and electrochemical impedance spectroscopy (EIS) analyses for the bioelectrochemical systems. The data were presented in Fig 3a,b and Supplementary Fig. 7a,b. The relevant results and methods were included in the revised manuscript (Line 179-192 and Line 572-578).

5. The authors stated that the battery is “self-sustaining and self-reproducing” and can even be used on Mars. However, the basic cycle stability tests are lacking in the manuscript. The systems collapsed only after 20 to 30 days of operation. In addition, the reason for these collapses and how to maintain a long-term stability should be discussed in the manuscript.

Response: Thank you very much for your comments. The statement of “self-sustaining and self-reproducing” represents a theoretical scenario. Since practically we have not achieved that, this statement was deleted in the revised manuscript.

The immediate cause of collapse for the system after about one month was due to cell death of cyanobacteria. In the batch experiment, the minimal nutrients such as inorganic ions over time became deficient, which induced the death of cyanobacteria. A possible strategy was continuously supplying of minimal inorganic nutrient for the system. The discussion on this aspect can be found in Line 401-404.

6. What was the power source utilized for light energy? Does the light used here mimic

the sunlight? Impacts of diurnal cycles?

Response: Thank you very much for your comments. The routine cultivation of cyanobacteria and the running of bio-solar cell were conducted in two independent illumination incubators, and the light sources are white fluorescence lamps (FLs) and white light-emitting diodes (LEDs), respectively. The spectrums of both light sources are close to the natural sunlight. The light source information was included in the Methods (Line 490-491 and Line 628-629).

In addition, during the revision of this manuscript, we supplemented the diurnal cycles (12 h light/ 12 h dark) experiments. The result was presented in Supplementary Fig. 17 and the relevant description of the result can be found in the main text (Line 297-304).

7. Line 375: Why was the M1-93 and rrB promoter chosen for the strain engineering?

Response: Thank you very much for your comment. The M1-93 and rrB are two strong constitutive promoters, which are widely used in *E. coli* for constitutive expression of genes. Thus, both promoters were chosen for constitutive expression of *cscB* and *gffA* genes in engineered strain of *E. coli* in this study. This was clarified in the revised manuscript (Line 442)

8. Some of the references used for the methods are not properly mentioned. For examples, lines 560, 469, 470 etc.

Response: Thank you very much for your comment. The citation style of these references has been corrected. Please refer to Line 469, 488, 508, 535, 536, 537 and 657. We also checked other references throughout the manuscript for consistency.

9. Figure S9: It is hard for the audience to visualize what the authors are pointing out. The electrographs look very similar to me and the presence of biofilms cannot be verified using these images.

Response: Thank you very much for your comment. We have supplemented two SEM images of bare RPPy and bare carbon cloth as controls in Supplementary Fig. 11 to visualize the biofilms formed on carbon fibers.

10. Figure S11: Is the transfer of lactate and sucrose a limiting factor in the design of the hydrogel battery?

Response: Thank you very much for your comment. As shown in Supplementary Fig. 13a, the mass-transfer efficiency of conductive hydrogel was not high, thus it might become a limiting factor for power density output of the bio-solar cell. Moreover, the overshoot phenomenon in polarization curve (Fig. 6c) might also be an indication of mass-transfer limitation of sucrose, lactate, and acetate in conductive hydrogel. A possible solution for that is decreasing its thickness but enlarging the contact area with cyanobacteria layer. Relevant description can be found in Line 308-313.

11. The introduction might be slightly misleading. While organic material transfer to heterotrophic organism is mentioned, it is unclear what are the organisms. For example, primary producers (carbon dioxide fixers by phototrophs and chemolithoautotrophs) are commonly predated by other planktons (metazoans like fish larvae and crustaceans) that drives the biological pump that ultimately stores carbon in the deeper ocean (current introduction seems to imply most of biogeochemical cycles only involve microbes). Hence, I think this portion should be clarified, and then focusing “ocean-battery” that is driven by microbial loop.

Response: Thank you very much for your comments. Yes, the “ocean-battery” concept should be confined to the microbial loop, thus we amended the relevant descriptions throughout the manuscript (Line 40-42, Line 45-46).

12. Line 58: Might want to take note of mentioning of methane production through decomposition and methanogenesis, especially in the deep ocean.

Response: Thank you very much for your comments. Methanogenesis is actually an important step in anaerobic degradation of organic carbon. It was mentioned in the Introduction of the revised manuscript (Line 54-56).

13. Line 88: Incomplete sentence.

Response: Thank you very much for your comment. This sentence has been simplified as “Using conductive hydrogel as a sediment-like matrix, we fabricate a miniaturized bionic ocean-battery with marine microbial ecological structure, which can stably convert

light into electricity". Please refer to Line 92-94.

Reviewer #3 (Remarks to the Author):

The present study describes the construction and operation of a novel bio-solar cell consisting of a four-species microbial community. This novel system mimics the basic ecological structure of marine ecosystems. The authors claimed that a three-dimensional electrode (carbonized conductive polymer polypyrrole) colonized with the four-species microbial community generated a maximum power density of $1.7 \text{ W}\cdot\text{m}^{-2}$ from light.

Furthermore, the authors also described the construction and operation of a miniaturized ocean-battery, making the claim that this battery directly converts light into electricity with a maximum power of $860 \mu\text{W}$.

The concept of "Marine ecosystems as rechargeable ocean-battery" as well as the term of "ocean-battery" are very interesting, I am complimenting with the authors for introducing these new ideas and having such a clear and well-made graphical materials (e.g., figure 1, 6 and 7).

In my view the present study deserves to be published in Nature Communications if the claim of "power output from light" is proven. This can be done by:

- 1) running the system operated with C+S2+S1+G in the dark and doing a comparison with an equal system operated under light condition;
- 2) running the system operated with C+S2+S1+G in the light after having consumed all the metabolite(s) introduced with the media used for growing the microorganisms (e.g., LB medium is rich in organics).
- 3) running the system operated with S2+S1+G (without cyanobacteria) under light condition and doing a comparison with an equal system operated with C+S2+S1+G (with cyanobacteria) under light condition;
- 4) calculating the power density by using the geometrical area receiving the light, not the geometric area of the anode (more details in my comment 6).

Response: Thank you very much for your comments and constructive suggestions. Your suggested control experiments are indeed very important for proving our claim. Following your suggestions, we supplemented the control experiments in the conductive hydrogel-based setups to prove the claim of power output from light.

Specifically, the control experiments (including dark control, cyanobacteria-free control and *E. coli*-free control) were performed in the conductive hydrogel-based systems. As shown in Fig. 6b, all the systems (including experimental group) generated an electrical current peak during the early two days, which could be ascribed to the residual metabolites or a bit of organic matter introduced by the medium. Nevertheless, all the electrical currents of three control groups decreased after two days, whereas it increased to a higher level for the experimental group due to the photosynthetic sucrose production. In addition, the photo-response of electrical current was observed under light/dark cycles (Supplementary Fig. 17). All these results proved that the power output was from light. The detailed descriptions can be found in Line 291-306.

Furthermore, the calculation of current densities produced by conductive hydrogel-based systems was normalized to the geometrical area of receiving the light (60 cm²), which was indicated in the Methods (Line 630-632, Line 648-649) and relevant figure legends (Line 911-912 in the main text, Line 161-162 in the Supplementary Information).

It should be pointed out that when we started to perform the above control experiments, we found a problem of a small amount of sugars contained in commercial carrageenan which was previously used to prepare the conductive hydrogel. To eliminate the interference caused by the impurity of carrageenan, we attempted to search for a replacement. Eventually, we found that agarose can also be used to prepare hydrogel by mixing with gelatin, and the newly prepared hydrogel did not contain any sugars or other nutrients that can be metabolized by microbial communities. Moreover, we found the newly prepared agarose-gelatin hydrogel was better than carrageenan-gelatin hydrogel in terms of mechanical strength (or hydrogel structure stability), thus the current output obtained using this newly prepared hydrogel was more stable and longer than previous one. For this newly prepared conductive hydrogel, we conducted the characterization experiments again and updated the results in Supplementary Fig. 13. The details of preparing the agarose-gelatin conductive hydrogel were described in the Methods (Line 606-612).

Additional comments

5) Page 7 and 8, line 129 - 134

The authors have defined the carbonizing conductive polymer polypyrrole as an 3D electrode. In comparison they have labeled the carbon cloth as “planar” (therefore not

3D).

However, by looking at the Supplementary Fig. 9, both electrodes seemed to me very similar in term of “porosity”/morphology. Could the authors provide any analytical quantification of the electrodes’ porosity/morphology? If not, both electrodes have to be named either 3D or planar.

Also, by looking at the Supplementary Fig. 4, the electrical output (as Coulomb) of RPPy looks to me very similar to the electrical output of CC. Therefore, unless the authors can prove a statistical difference, those two electrodes displayed an equivalent electrical output. Could the authors comment on this?

Response: Thank you very much for your comments. In consideration of the similarity in roughness and carbon-based materials, both electrodes were termed as “porous electrode” and no longer named 3D or planar in the revised manuscript. The major difference between them was thickness, and we supplemented the actual photographs of both electrodes in Supplementary Fig. 4 for clarification.

6) Abstract and main text

In the abstract, and various parts of the main text, the authors have claimed “the synthetic community inside a three-dimensional electrode generated a maximum power density of $1.7 \text{ W}\cdot\text{m}^{-2}$ from light”.

According with the Method (line 499-501), this was calculated according to Ohm's law and normalized to the geometric area of the anode (6.25 cm^2).

This is in my view not correct. As the authors claim “from light”, the normalization has to be based on the geometrical area receiving the light, not the geometric area of the anode.

Response: Thank you very much for your comments. In the conductive hydrogel-based systems, the electrochemical setups were illuminated by light directly. In this case, the calculation of current/power density was normalized to the geometrical area of receiving the light, which is in line with your suggestions. Please refer to Fig. 6 and Supplementary Fig. 17.

For the synthetic microbial communities attached to RPPy and carbon cloth, the electrochemical setups were not illuminated at the discharging stage. In other words, to avoid the disturbance of oxygen produced by cyanobacteria to the anodic biofilm, the synthetic communities were constructed using a mode of temporal separation organization, in which the charging process of photosynthetic sucrose production and

the discharging process of anaerobic sucrose oxidation were implemented sequentially under light and dark conditions, respectively.

Therefore, we calculated the current densities in this case by normalizing to the anode area, which were adopted more commonly in bioelectrochemical fields including bio-solar cells. In order to avoid confusion, the relevant statements such as “from light” and “directly from light” were excluded when describing these results. The calculation details were indicated in the legends of relevant figures where current/power densities were presented (Line 854-855, 869-870 in the main text and Line 81-82, 90-91 in the Supplementary Information).

7) More details of the electrochemical apparatus are required.

Please provide in the main text more details for the electrochemical apparatus used for doing the experimental work. For example, actual dimensions and geometry. I suggest to provide a schematic as well as an actual photo of the apparatus (those could be displayed in the supplementary material).

Response: Thank you very much for your suggestions. We have supplemented the schematic diagrams and photographs of the electrochemical apparatus used in this study in Supplementary Fig. 19.

8) The overall energy balance.

As the light intensity, amount of metabolites and electrical output are known, please provide the overall energy balance of the presented system.

Please start from light (W/m^2) going into sucrose ($mmol/m^2$) and then acetate/acetate ($mmol/m^2$) for then finishing with current (mW/m^2).

Response: Thank you very much for your comment. The overall energy balance of the final conductive hydrogel-based system was shown in Fig. 6d. The energy flux (mW/m^2) and energy efficiency (%) of light-to-sucrose, sucrose-to-lactate, lactate-to-electricity, and light-to-electricity were calculated. The relevant descriptions can be found in Line 313-317.

9) Figure 6d

In the figure 6d, the authors presented the performance comparison of this study with previously demonstrated bio-solar cells, showing how their new system exceeded all the

previously tested. This comparison is not very informative as the data could come from devices with a very different size. Please change this as shown in the Supplementary figure 7 (power density).

Response: Thank you very much for your comment. The previous Fig. 6d has been moved into the Supplementary Fig. 9b in the revised manuscript.

10) Was the light ON?

The experimental data used to generate the data shown in figure 2d-g, 5b and 6b were obtained with light ON or OFF? Please also indicate the light intensity and quality. Also, please provide results (e.g., chronoamperometry) showing few dark/light cycle.

Response: Thank you very much for your comments. The data shown in Fig. 2d-g were obtained indirectly from light, in which the temporal separation organization was adopted, i.e. photosynthetic sucrose production was performed in the light and anaerobic sucrose oxidation was performed in the dark. The data shown in Fig. 5b was the current produced by *Shewanella* by oxidizing lactate with light OFF. The data shown in Fig. 6b were obtained with light ON.

The light resources and intensities were indicated in the relevant figure legends (Line 906-907) and Methods (Line 490-491 and Line 628-629).

Following your instruction, we supplemented the experiment of performing conductive hydrogel-based setups under light/dark cycles (12 h/12 h). The data was shown in Supplementary Fig. 17, and the relevant descriptions can be found in Line 297-304.

Dr. Paolo Bombelli.

Reviewers' Comments:

Reviewer #1:

Remarks to the Author:

Thank you for the corrections, the comments and the changes made.

Reviewer #2:

Remarks to the Author:

The authors have addressed my comments reasonably well. I do not have further comments.

Reviewer #3:

Remarks to the Author:

The manuscript has been substantially improved compared to the previous submission.

However, the process of revision has also led to tame some of the claims and so the significance to the field. For example, in the previous submission the authors have written in the abstract "This battery directly converts light into electricity with a maximum power of 860 μ W, exceeding the performance of previously demonstrated bio-solar cells."

By contrast, in the current one, the authors have re-written the abstract with a more modest claim "This battery directly converts light into electricity with a maximum power output of 380 μ W and stably operates for over one month."

Overall, on balance my recommendation is to publish this manuscript in Nature Communication as the work is original and well presented, the conclusions are supported by the data and the methodology sounds.

Dr. P. Bombelli.

REVIEWER COMMENTS

Reviewer #1 (Remarks to the Author):

Thank you for the corrections, the comments and the changes made.

Response: Thank you very much for your positive comments.

Reviewer #2 (Remarks to the Author):

The authors have addressed my comments reasonably well. I do not have further comments.

Response: Thank you very much for your positive comments.

Reviewer #3 (Remarks to the Author):

The manuscript has been substantially improved compared the previous submission. However, the process of revision has also led to tame some of the claims and so the significance to the field. For example, in the previous submission the authors have written in the abstract “This battery directly converts light into electricity with a maximum power of 860 μ W, exceeding the performance of previously demonstrated bio-solar cells.”

By contrast, in the current one, the authors have re-written the abstract with a more modest claim “This battery directly converts light into electricity with a maximum power output of 380 μ W and stably operates for over one month.”

Overall, on balance my recommendation is to publish this manuscript in Nature Communication as the work is original and well presented, the conclusions are supported by the data and the methodology sounds.

Dr. P. Bombelli.

Response: Thank you very much for your positive comments. We are very grateful to your constructive suggestions and the supplemented experiments that you suggested further strengthened this research.